# TERRA recruitment of polycomb to telomeres is essential for histone trymethylation marks at telomeric heterochromatin

Juan J. Montero[1], Isabel López-Silanes[1], Diego Megías[2], Mario F. Fraga[3], Álvaro Castells-García[4,5] & Maria A. Blasco[1]

TERRAs are long non-coding RNAs generated from the telomeres. Lack of TERRA knockout models has hampered understanding TERRAs' functions. We recently identified chromosome 20q as one of the main origins of human TERRAs, allowing us to generate the first 20q-TERRA knockout models and to demonstrate that TERRAs are essential for telomere length maintenance and protection. Here, we use ALT 20q-TERRA knockout cells to address a direct role of TERRAs in telomeric heterochromatin formation. We find that 20q-TERRAs are essential for the establishment of H3K9me3, H4K20me3, and H3K27me3 heterochromatin marks at telomeres. At the mechanistic level, we find that TERRAs bind to PRC2, responsible for catalyzing H3K27 tri-methylation, and that its localization to telomeres is TERRA-dependent. We further demonstrate that PRC2-dependent H3K27me3 at telomeres is required for the establishment of H3K9me3, H4K20me3, and HP1 binding at telomeres. Together, these findings demonstrate an important role for TERRAs in telomeric hetero-chromatin assembly.

[1] Telomeres and Telomerase Group, Molecular Oncology Program, Melchor Fernández Almagro 3, E-28029 Madrid, Spain. [2] Confocal Microscopy Unit, Spanish National Cancer Centre (CNIO), Melchor Fernández Almagro 3, E-28029 Madrid, Spain. [3] Cancer Epigenetics Laboratory, Nanomaterials and Nanotechnology Research Center (CINN-CSIC)-Universidad de Oviedo, Institute of Oncology of Asturias (IUOPA) and Instituto de Investigación Sanitaria del Principado de Asturias (ISPA), Avda De la vega, 4-6, 33940 El Entrego, Spain. [4] Centre for Genomic Regulation (CRG), The Barcelona Institute of Science and Technology, Drive Aiguader 88, 08003 Barcelona, Spain. [5] Universitat Pompeu Fabra (UPF), 08003 Barcelona, Spain. Correspondence and requests for materials should be addressed to M.A.B. (email: mblasco@cnio.es)

Telomeres are nucleoprotein structures at the ends of chromosomes that protect them from being recognized as DNA double-strand breaks, thus preventing chromosomal end-to-end fusions[1]. In vertebrates, telomeres consist of tandem repeats of the TTAGGG sequence bound by the so-called telomere-binding proteins or shelterin, which are essential for the formation of a functional telomere cap[2]. Telomere repeats can be generated de novo by telomerase, a reverse transcriptase (TERT) that elongates chromosome ends by using an RNA component (TERC) as template[1]. In mice and humans, telomerase is highly expressed in embryonic pluripotent stem cells, and this expression is downregulated after birth leading to progressive telomere erosion with aging owing to the incomplete replication of linear chromosomes[3,4]. In contrast, cancer cells aberrantly overexpress telomerase allowing for the ability of cancer cells to proliferate indefinitely[5]. Cancer cells can also use an independent mechanism to elongate telomeres known as alternative lengthening of telomeres (ALT), that is based on homologous recombination between telomeric sequences[6].

Interestingly, both telomere length and telomere recombination are subjected to a higher-order regulation involving epigenetic modifications of the telomeric chromatin[7]. In particular, mammalian telomeres are enriched in heterochromatic marks, including HP1 binding, H3K9 and H4K20 tri-methylation histone marks, as well as hypermethylation of subtelomeric DNA[7–13]. These marks have been proposed to negatively regulate telomere length and telomere recombination[7]. In particular, cells lacking the histone methyltransferases Suv39 or Suv420, as well as cells lacking the DNA methyltransferases DNMT1 and 3 have markedly elongated telomeres[9,11,12]. Thus, disruption of this silent chromatin environment results in loss of telomere-length control and in increased telomere recombination. In turn, our group also showed that progressive telomere loss associated to cell division reduces chromatin compactation at telomeric and subtelomeric domains, which may favor telomere-elongation mechanisms[7,8,13]. Telomere chromatin also shows a decrease of heterochromatic histone marks during nuclear reprogramming[14], coinciding with net telomere elongation by telomerase[14].

Despite the heterochromatic environment of the telomere, telomeres are transcribed giving rise to long non-coding UUAGGG-repeat transcripts known as Telomeric repeat-containing RNAs (TERRAs) or TelRNAs[15,16]. TERRAs are transcribed from the subtelomere toward the telomere and show a spotted nuclear pattern as detected by RNA-FISH[15,16]. A proportion of TERRAs spots colocalizes with telomeres, suggesting that TERRAs are part of the telomeric chromatin[15,16]. At telomeres, TERRAs are essential to maintain telomere length and telomere protection[17–20], and therefore can be considered as bone fide telomere components. TERRAs have been also recently found to bind to extratelomeric regions, explaining the non-telomeric TERRA spots[17].

TERRAs have been implicated in telomere protection[17–21], heterochromatin formation[21], telomere replication[22,23], and in telomere elongation by homologous recombination through the formation of DNA-TERRA hybrids[24–26]. However, understanding of TERRA in vivo function and the direct involvement of TERRA in all the above-mentioned processes is largely pending owing to lack of loss-of-function TERRA models. In this regard, we recently identified the mouse and the human TERRA locus[18,20]. We found that in both human and mouse, TERRAs do not arise from all chromosomes but from a few of them[18,20]. In human cells, the majority of TERRAs arise from a single locus in chromosome 20q[20]. This allowed us to first generate knockout cells for this locus, which showed markedly reduced TERRA levels, thus demonstrating that 20q is the main origin of human TERRAs. 20q-TERRA knockout cells also demonstrated that

TERRAs are essential for telomere maintenance and telomere protection[20]. Here, we use this human TERRA loss of function model to assess a direct role of TERRA long non-coding RNAs in telomeric heterochromatin formation, as well as unveil the underlying molecular mechanisms.

In particular, there is mounting evidence that non-coding RNAs are involved in heterochromatin formation across species both through cis and trans mechanisms. This includes dosage compensation in mammals, imprinting, and demarcation of gene-silencing chromosomal domains[27,28]. In addition, non-coding RNAs transcribed at centromeres are also proposed to be involved in higher-order chromatin structures, and their transcription to be important for the deposition of CenH3 (homologous centromere-specific histone H3 variants; ref. [29]).

In the case of TERRAs, not only do they colocalize with telomeric chromatin in cis and trans[15,16,18], but they can also form DNA–RNA hybrids or R-loops at the telomere[24–26], a type of structure shown to epigenetically modify the genome[30,31]. TERRAs have also been shown to interact with HP1 and H3K9me3[21]. Intriguingly, TERRAs have been proposed to correlate with H3K27 deposition at extratelomeric sites, a histone mark placed by EZH2, the catalytic subunit of the Polycomb repressive complex 2 (PRC2). However, whether TERRAs regulate H3K27 deposition at telomeres or, more generally, whether TERRAs have a direct role in the establishment of telomeric chromatin status is largely unknown owing to the lack of KO models for TERRAs.

Here, we set to address a direct involvement of TERRAs in the regulation of a higher-order regulation of chromatin at telomeres by using a panel of human 20q-TERRA KO cells with markedly reduced TERRA levels. We find that depletion of TERRAs in human U2OS cells cause a strong loss of H3K9me3 and H4K20me3 tri-methylation histone marks at telomeric chromatin, demonstrating direct involvement of TERRAs in the establishment of telomeric heterochromatin. In addition, we make the finding that human telomeres are enriched for the H3K27m3 tri-methylation facultative heterochromatin mark, which is catalyzed by the PRC2 complex, and that this mark is also decreased in 20q-TERRA KO cells. Indeed, we describe here that TERRAs directly bind the PRC2 complex components EZH2 and SUZ12 and that this binding is critical for PRC2 recruitment to telomeres. We further demonstrate that PRC2 at telomeres is required for the establishment of H3K27me3, H3K9me3, H4K20me3, and HP1 binding at telomeres. In summary, we describe for the first time a role for PRC2 in the establishment of telomeric chromatin, which is regulated by TERRAs, thus demonstrating an important role for TERRAs in telomeric heterochromatin assembly.

## Results

**Generation of a panel of 20q-TERRA KO cells.** We first generated a large panel of human U2OS cells KO for the 20q-TERRA locus with the CRISPR-Cas9 system using the same strategy described by us[20]. To increase the efficacy of deletion we used an all-in-one plasmid system[32], which contains the two gRNAs (E1 and S2) under the U6 promoter, the Cas9 protein, and a GFP cassette (see Materials and Methods; Fig.1a). EGFP-positive cells were sorted by flow cytometry and after PCR screening, we obtained eight clones that carried the deletion in homozygosis (clones #A7, E1, E6, E8, and H8; Fig. 1b). As controls, U2OS cells were electroporated with the same plasmid but without gRNAs (control clones #1 and #2). As expected, deletion of the 20q-TERRA locus resulted in a significant reduction in total TERRA levels in the deleted clones (A7, E1, E6, E8, and H8) compared with the original U2OS cellular pool and with the WT clones (#1 and 2; Fig. 1c), as determined by dot-blot analysis. Note that,

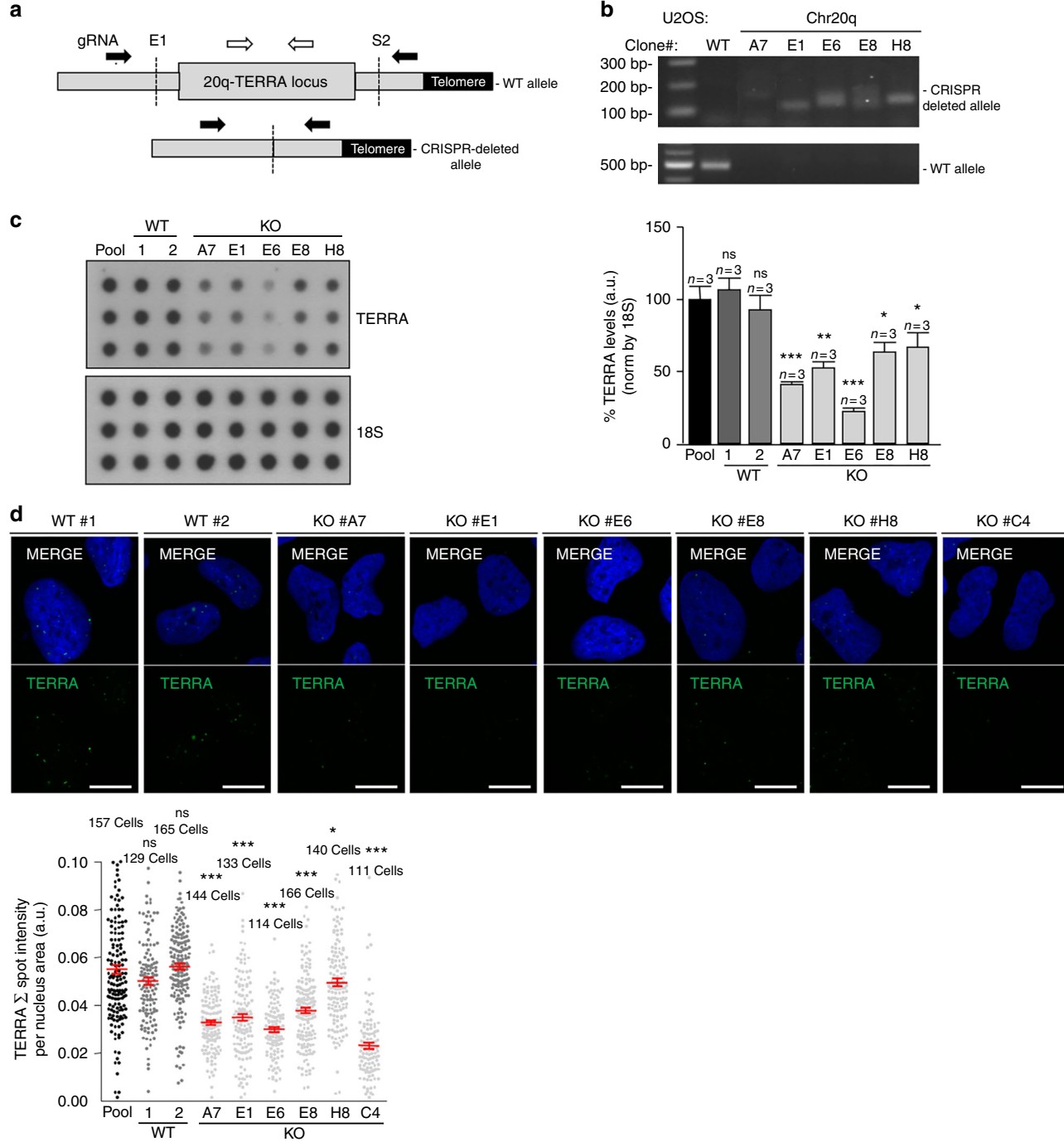

**Fig. 1** Deletion of the TERRA-20q locus markedly affects TERRA expression. **a** Scheme depicting the WT and the CRISPR-deleted allele for the 20q-TERRA locus located in the subtelomere of the chromosome 20, q-arm. The position of the gRNAs (E1 and S2) and the primers used to genotype the deletions are also shown. The black arrows represent the primers to amplify the CRISPR-deleted allele and the white arrows the ones to amplify the WT allele located inside the 20q-TERRA locus. **b** Ethidium bromide gels showing the WT and the CRISPR-deleted allele for the 20q-TERRA locus detected by PCR in a WT cellular pool and in different clones of the U2OS cells. **c** RNA from the a WT cellular pool, WT expanded clones (#1 and 2), or 20q-TERRA KO clones (#A7, E1, E6, E8, and H8) from the USOS cell line was isolated and used for TERRA detection by RNA dot-blot with a probe against the TERRA-UUAGGG-tract; 18S serves as loading control. (Graph) TERRA quantification normalized by 18S (mean values ± s.e.m., $n = 3$ biological replicates). **d** Representative confocal microscopy images of RNA-FISH against TERRA-UUAGGG-tract (green) in the U2OS WT clones (#1 and 2) and in the 20q-TERRA KO clones (#A7, E1, E6, E8, H8, and C4). Scale bar, 10 μm. (Graph) Quantification of the total spot intensity per nucleus normalized by nucleus area (mean values ± s. e.m., $n$ = cells analyzed). One-way ANOVA with Dunnett's post test was used for the statistical analysis (*$p < 0.05$, **$p < 0.01$, and ***$p < 0.001$)

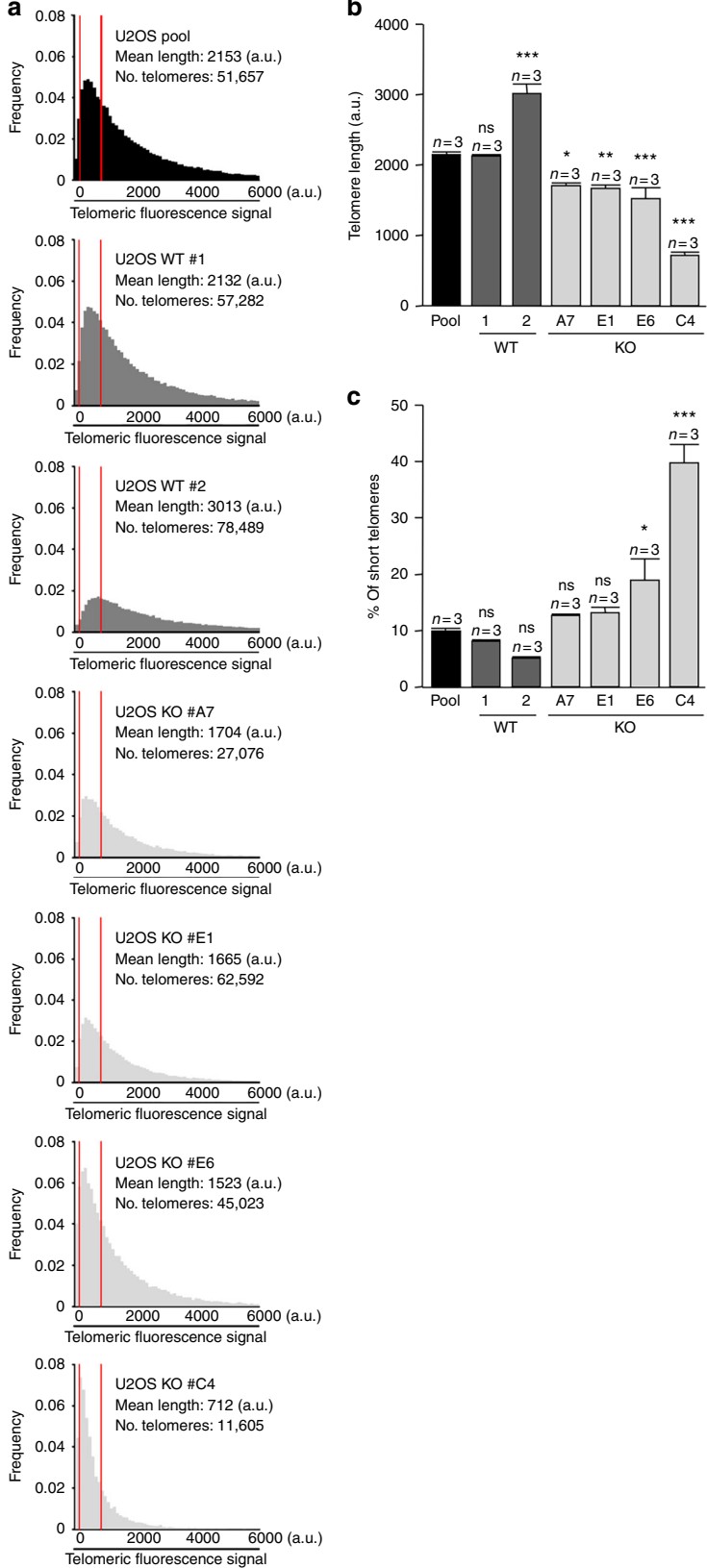

**Fig. 2** Deletion of the 20q-TERRA locus decreases telomere length. **a** Representative frequency graphs of telomere length distribution (a.u.) measured in the U2OS WT pool (black), in the WT clones (#1 and 2; dark gray), and in the 20q-TERRA KO clones (#A7, E1, E6, and C4; light gray). The mean telomere length and the number of telomeres analyzed is shown. The red lines are arbitrary lines placed in the exact same position in each frequency graph to visualize differences between samples. **b** Graph showing the quantification of the mean telomere length in the U2OS cells WT pool, the WT expanded clones (#1 and 2), and in the 20q-TERRA KO clones (#A7, E1, E6, and C4) by HT-Q-FISH (mean values ± s.e.m., $n$ = technical replicates). **c** Graph showing the percentage of short telomeres in the same settings. Short telomeres are considered those in the 10% percentile of the total telomere length distribution (mean values ± s.e.m., $n$ = technical replicates). One-way ANOVA with Dunnett's post test was used for the statistical analysis (*$p < 0.05$, **$p < 0.01$, and ***$p < 0.001$)

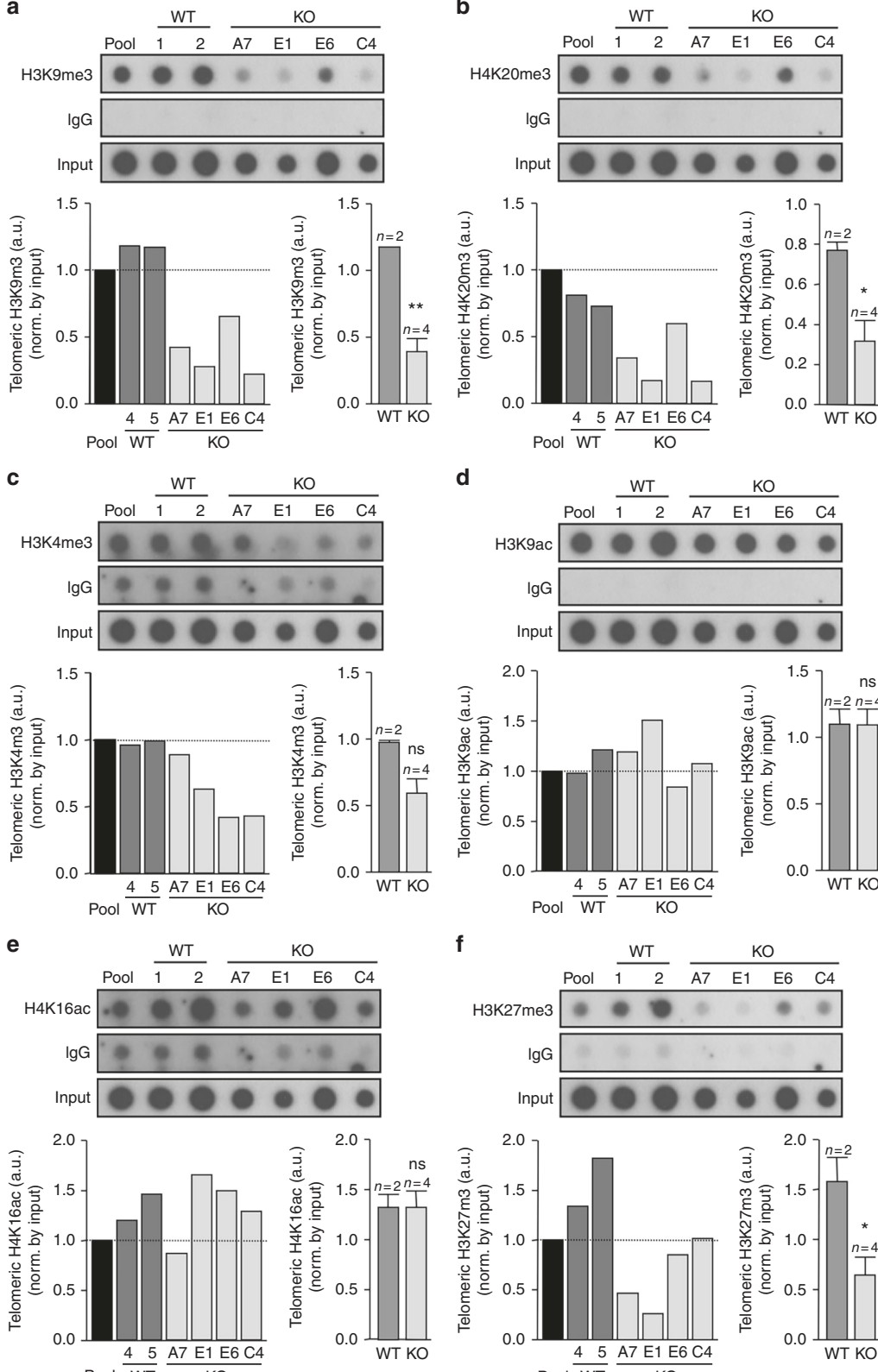

**Fig. 3** TERRAs are essential for the assembly of heterochromatic histone marks at the telomere, including H3K27me3. **a** ChIP-dot-blot of the H3K9m3, **b** H4K20m3, **c** H3K4m3, **d** H3K9ac, **e** H4K16ac, and **f** H3K27me3 histone marks for the U2OS cells WT pool, WT clones (#1 and 2), and from the 20q-TERRA KO clones (#A7, E1, E6, and C4), hybridized with a southern probe against the telomeric repeat. ChIP-dot-blot for IgG was used as a control. IgG ChIP-dot-blot shared by different antibodies shows different exposure times according to the best exposure time required for each antibody. DNA input signal is also shown. Below the ChIP-dot-blot for each mark is shown the quantification of the immunoprecipitated telomeric repeats normalized by the input for each individual sample (left graph) and for all WT clones vs. the 20q-TERRA KO clones (right graph) (mean values ± s.e.m., $n$ = independent clone). Student's $t$-test was used for the statistical analysis (*$p < 0.05$, **$p < 0.01$, and ***$p < 0.001$)

since the 20q-TERRA locus is an important TERRA locus but not the only one[20], we do not expect the complete abolition of TERRA expression in the 20q-TERRA KO clones. Moreover, the fact that in some of the 20q-TERRA KO clones the TERRA downregulation is only 20–50% might be related to (1) the adaptation to the cell culture conditions during clonal cell expansion and (2) the compensation of other loci, for example, the Xp[20] locus or others. Nevertheless, 20q-TERRA is the only locus in which a formal demonstration of its TERRA genuineness has been carried out by genetic means so far[20]. We confirmed TERRA's downregulation by RNA-FISH using probes to detect

the TERRA's -UUAGGG-repeat. In particular, we found a significant 20–50% reduction in total TERRA levels in the 20q-TERRA KO clones compared to the controls (Fig. 1d). The 20q-TERRA KO C4 clone from our previous work[20] was included in this analysis in parallel with the newly generated clones (Fig. 1d).

Next, we set to confirm whether the newly generated 20q-TERRA KO clones also showed a telomere-shortening phenotype as previously described[20]. To this end, we performed telomeric quantitative FISH (Q-FISH) to quantify telomere fluorescence and also included formerly generated 20q-TERRA KO clone C4 as control[20]. As shown in Fig. 2, deletion of the 20q-TERRA locus

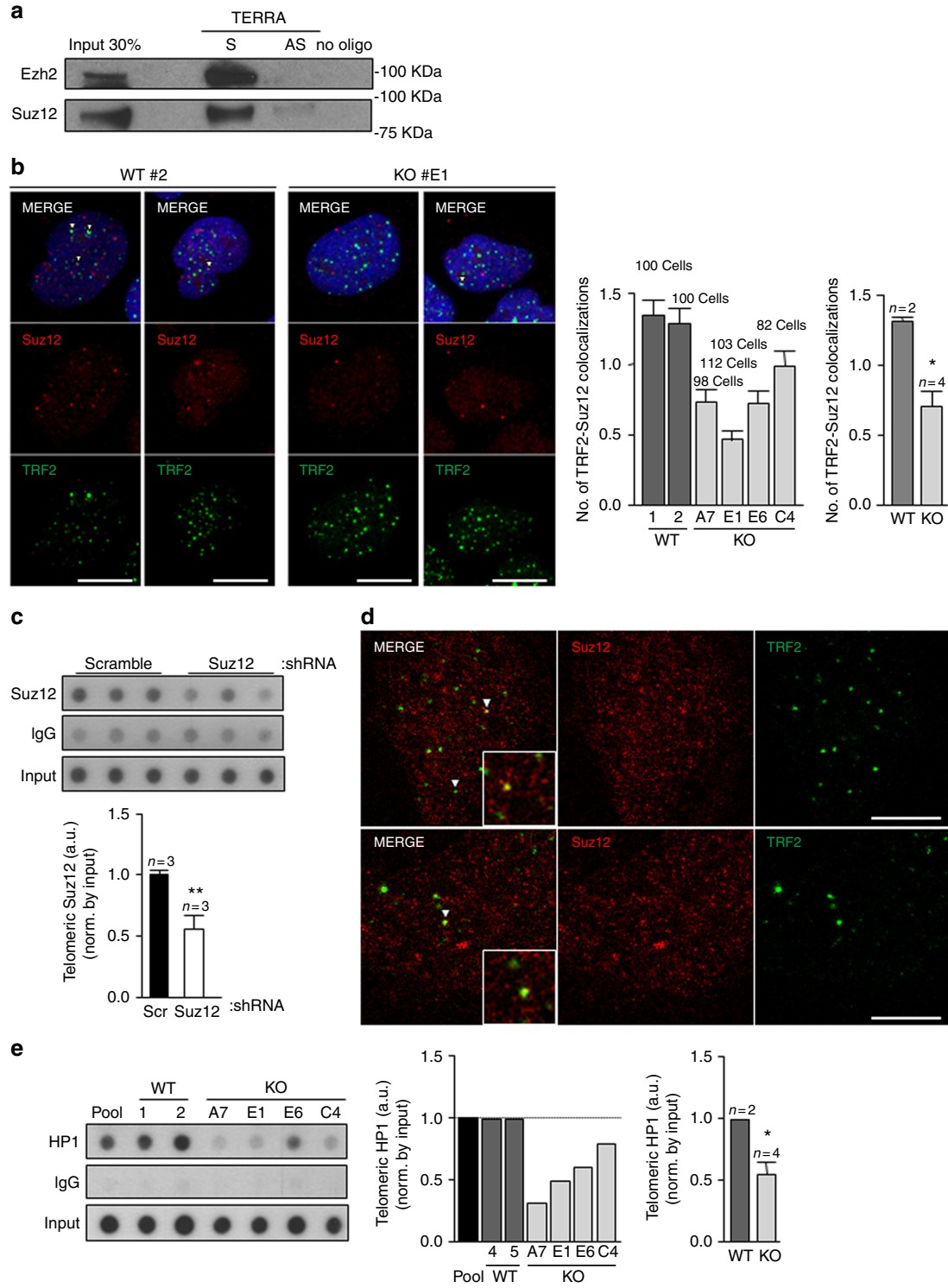

in the new clones and in the C4 clone resulted in a marked loss of telomeric sequences as seen in the switch of the distribution of telomere length frequencies toward lower values (Fig. 2a) and in the significant decrease in telomere fluorescence intensity (Fig. 2b). This was concomitant with a significant increase in the percentage of very short telomeres as determined by low telomere fluorescence (short telomeres considered those in the 10th percentile of the total telomere length distribution; Fig. 2c). We observed no differences between the pool and the WT clone #1 and an increase in telomeric length between the pool and the WT clone #2 (Fig. 2b) that could be explained as a phenomenon related to cell expansion.

In summary, the new 20q-TERRA KO clones generated here reproducibly show marked TERRA downregulation and decreased telomere length, thus supporting that the 20q locus is a bona fide TERRA origin in human U2OS cells.

**TERRAs' assembly heterochromatic histone marks at telomeres**. To address the role of TERRAs in telomeric chromatin formation, we evaluated the impact of 20q-TERRA deletion on the abundance of different histone marks at telomeres. To this end, we used chromatin immunoprecipitation (ChIP), followed by dot-blot (Methods). We first confirmed the presence of heterochromatic histone marks previously shown by us to be enriched at telomeres, namely tri-methylated H3K9me3 and H4K20me3[11,12] (Fig. 3a,b). Interestingly, we found that TERRA downregulation in the different 20q-TERRA KO clones resulted in a significant decrease in abundance of H3K9m3 and H4K20me3 heterochromatic marks at telomeres compared to the WT pool and WT clones (Fig. 3a,b). These findings indicate a role for TERRAs in the establishment of these telomeric heterochromatin marks. Note that the apparent lack of "dose–response" between TERRA levels and heterochromatic marks at telomeres might be related to the clonal expansion and growth adaptation of the U2OS tumoral cell line.

Next, we studied the abundance of active chromatin histone marks at telomeric chromatin, such as H3K4m3, as well as histone acetylation H3K9ac and H4K16ac marks. However, we did not find that TERRA levels significantly affected any of these active chromatin marks (Fig. 3c–e), indicating that TERRAs do not regulate active transcription histone marks at telomeres.

**TERRAs are essential for deposition of H3K27me3 at telomeres**. The facultative heterochromatin mark, H3K27me3, is established by the polycomb complex 2 and has been recently shown to cooperate with the heterochromatic histone mark H3K9me3 to recruit HP1 to heterochromatin[33]. In spite of the fact that telomeres are enriched in both HP1 and H3K9me3, a role for H3K27me3 at telomeres has not been explored before. Here, we set to address whether H3K27me3 is present at human telomeres and whether the abundance of this mark is influenced

by TERRA levels. Interestingly, we found that TERRA downregulation in the different 20q-TERRA KO clones resulted in a significant decrease in abundance of H3K27me3 marks at telomeres compared to the WT pool and WT clones (Fig. 3f), demonstrating that TERRAs regulate the assembly of H3K27me3 at telomeres. The fact that all 20q-KO clones undergo similar changes at these heterochromatic marks, similar changes in TERRA levels, and have the same telomeric phenotype (see above) supports that the changes observed are not due to the accumulation of mutations during cell culture, to the monoclonal expansion, or to the presence of CRISPR off-targets. Together, these findings show that TERRAs are needed for the assembly of H3K9me3, H4K20me3, and H3K27me3 histone chromatin marks at telomeres, thus demonstrating a role for TERRAs in the establishment of telomeric heterochromatin.

**TERRAs bind PRC2 and regulates its recruitment to telomeres**. The finding of TERRA-dependent assembly of H3K27m3 at telomeres prompted us to investigate whether TERRAs directly regulate the presence at telomeric chromatin of the polycomb PRC2 complex responsible for catalyzing the methylation of H3K27[34]. First, we set to address a direct interaction between TERRAs and PRC2 components. To this end, we performed a TERRA biotin pull-down assay followed by western blot to detect PRC2 components. For specific TERRA binding, we incubated U2OS cells' nuclear extracts with a biotinylated oligo consisting of eight TERRA-UUAGGG repeats (8 × UUAGGG; Methods). As a negative control, we used a biotinylated oligo consisting of eight CCCUAA repeats (8 × CCCUAA; the TERRA antisense sequence), as well as beads alone. To detect presence of the PRC2 complex in the pull-down we used antibodies against two of core proteins of the PRC2 complex, EZH2 and SUZ12. Importantly, we found that both EZH2 and SUZ12 were able to specifically bind to the sense TERRA oligo ("S" in Fig. 4a) but not to the control antisense oligo ("AS" in Fig. 4a). Similarly, we could not detect EZH2 and SUZ12 when only the beads were incubated with the cell extracts ("no oligo" in Fig. 4a). These findings demonstrate that the PRC2 complex specifically interacts with TERRA RNAs. During the preparation of this manuscript, the TERRA–PRC2 interaction was also proven by electrophoretic mobility shift assay using a recombinant holo-PRC2 5-mer complex (EZH2, EED, SUZ12, RBBP4, and AEBP2) and a TERRA oligo[35]. The authors also found that PRC2 has a general affinity for G-rich RNA especially those capable of folding into G-quadruplexes[35]. Later on, Chu and co-workers demonstrated the direct interaction between TERRAs and EZH2 by iDRIP (identification of direct RNA-interacting proteins)[17], which reinforces our findings.

We next set to address whether PRC2 is specifically located at telomeres and whether TERRAs are necessary for its telomeric location. To this end, we used immunofluorescence to evaluate the presence of the SUZ12 PRC2 component at telomeres. As

**Fig. 4** TERRAs bind PRC2 and modulates its own and HP1 recruitment to telomeres. **a** A TERRA biotinylated RNA oligo (S) was incubated with nuclear extracts from U2OS cells and their association with Ezh2 and SUZ12 was detected by western blotting. A biotinylated control RNA oligo corresponding to the complementary sequence (AS) of the same length as the biotinylated TERRA (N$_{48}$) was used as control. Biotin pull-down in the absence of RNA oligo (no oligo) was included to monitor inespecific binding to the beads. **b** Representative images of the average number of colocalizations found on double immunostaining to TRF2 (green) and SUZ12 (red) in the U2OS WT and 20q-KO clones. Arrowheads indicate colocalization events. Scale bar, 10 μm. (Left graph) Quantification of the colocalization in each of the WT and 20q-TERRA KO clones (mean values ± s.e.m., n = number of cells) and (right graph) in all WT vs. the 20q-TERRA KO clones (mean values ± s.e.m., n = independent clone). **c** Telomeric ChIP-dot-blot of SUZ12 in U2OS cells infected with scramble or SUZ12 shRNA. IgG was used as a control. DNA input is also shown. (Graph) Quantification of the signal from the immunoprecipitated telomeric repeats normalized by the input (mean values ± s.e.m., n = technical triplicates). **d** Representative confocal STED super-resolution images showing the colocalization between TRF2 (in green) and SUZ12 (in red) in U2OS cells. Zoom: a colocalization event. Scale bar 5 μm. **e** Telomeric ChIP-dot-blot for HP1 in WT and 20q-TERRA KO clones. IgG was used as a control. DNA input signal is also shown. (Left graph) Quantification of the signal from the immunoprecipitated telomeric repeats normalized by the input for each individual sample and (right graph) for all WT vs. 20q-TERRA KO clones (mean values ± s.e.m., n = independent clone). Student's t-test was used for statistical analysis (*p < 0.05 and **p < 0.01)

previously described[36], we observed that SUZ12 formed nuclear foci (red dots in Fig. 4b). Interestingly, we found that approximately 1.5 SUZ12 spots (in red) per nucleus colocalized with TRF2 (in green; Fig. 4b), thus indicating the presence of the SUZ12 PRC2 protein at telomeres. If we refer this quantification to the number of telomeres, 3.5% of the telomeres (as detected by

TRF2) colocalize with SUZ12. Importantly, SUZ12-TRF2 colocalization events were significantly decreased by ~50% in 20q-TERRA KO clones (#A7, #E1, #E6, and #C4) compared to the WT clones (#1 and #2), demonstrating that the SUZ12 location to telomeres was dependent on TERRAs (Fig. 4b). The specificity of this reproducible, although low-in-number interaction, of SUZ12

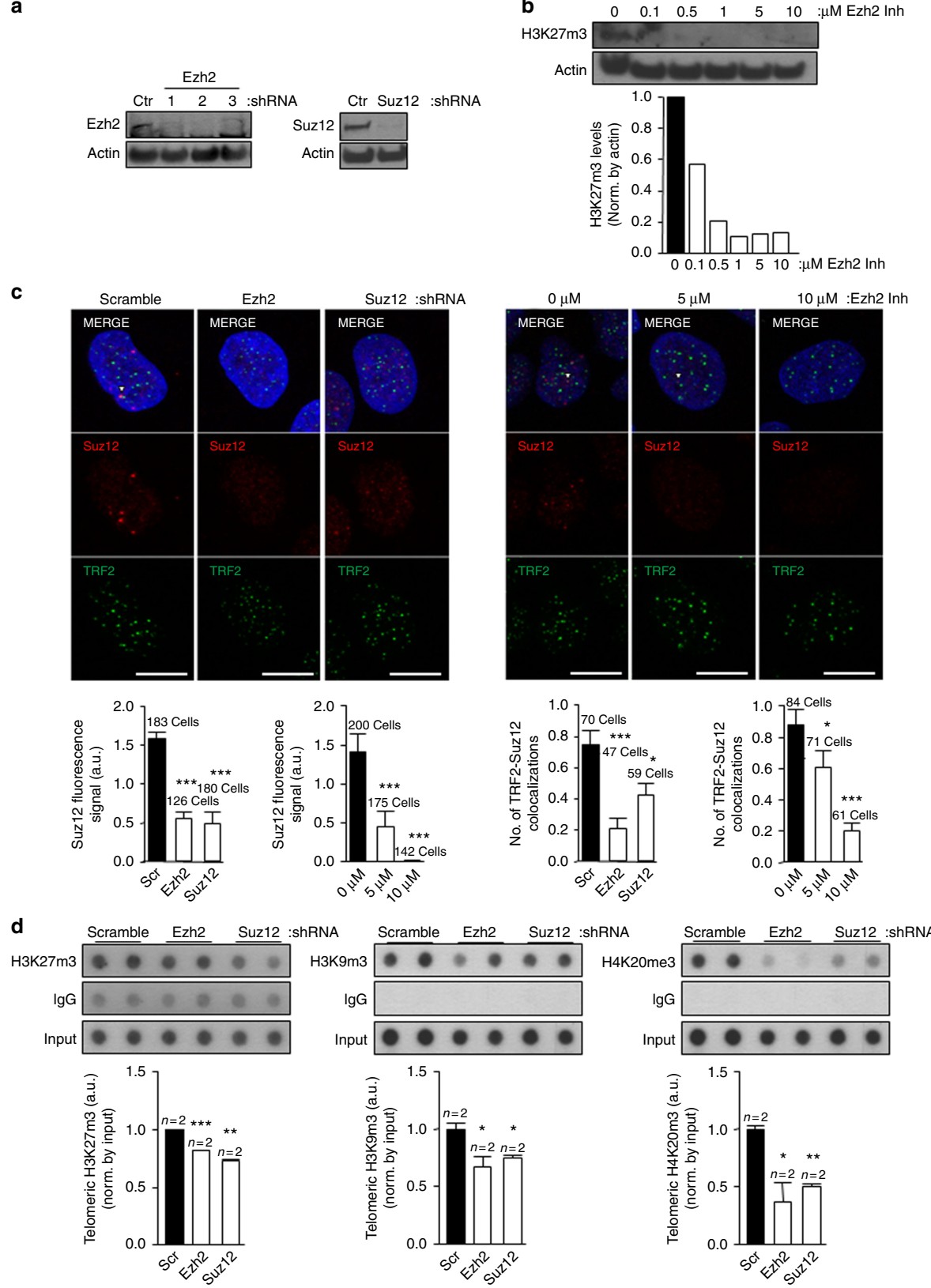

with the telomere was further confirmed by immunofluorescence upon downregulation of PCR2 components with shRNAs or with chemical inhibitors (see below). In addition, a SUZ12 ChIP followed by telomeric dot-blot confirmed this interaction of SUZ12 with telomeric DNA, which was decreased upon downregulation of SUZ12 levels by shRNA (Fig. 4c). The definitive proof of the interaction of SUZ12 with the telomere was achieved by super-resolution microscopy (Fig. 4d). As seen in the confocal super-resolution images, we found colocalization events between SUZ12 and the telomeric-binding protein TRF2 (Fig. 4d). Nevertheless, additional controls to prove this interaction were performed by running three different types of randomization approaches. First, we carried out a double immunofluorescence of SUZ12 and TRF2 run in parallel with one using an anti-centromere antibody (ACA) and TRF2. As it can be seen in Supplementary Fig. 1, the number of colocalizations between SUZ12 and TRF2 was significantly higher than the ones observed by chance between TRF2 and the ACA antibody. Second, using an interaction plugin on Fiji[37], the interaction potential between TRF2-SUZ12 was calculated. Interaction is defined as the spatial distribution of signal in one channel being not independent of the signal distribution in the other channel. For that, objects were identified and their NND (nearest neighbor distance) calculated, and compared with one probability density function of the NND if the signals were independent. For the Suz-TRF2 colocalizations the strength of the interaction is superior to zero, indicating that the spatial distribution of TRF2-SUZ12 is dependant (Supplementary Fig. 2). When compared against 10,000 Monte Carlo samples of NND distributions corresponding to the null hypothesis of "no interaction", the results are statistically significant ($p < 0001$). Third, we used the randomization tool of the Definiens Developer XD.2 software. In this way, TRF2 spots were digitally found and a randomization of the TRF2 signal was performed (16,000 randomized pictures; Supplementary Fig. 3A). The number of TRF2 spots found in four original pictures and in the ones randomized is shown in the Supplementary Fig. 3B (please, note that each picture contains different number of nuclei). Next, the colocalizations between TRF2 and SUZ12 were identified both in the original and in the randomized pictures (see Methods). Importantly, we found that the number of random colocalizations was significantly lower than in the original pictures (Supplementary Fig. 3C). The percentage of randomized pictures with ≥colocalizations is significantly lower than in the original pictures (Supplementary Fig. 3D). All together, these findings demonstrate that SUZ12 interacts with the telomere and that this interaction is TERRA-dependent.

**TERRAs are necessary for HP1 recruitment to telomeres.** We previously showed that mammalian telomeres are enriched in HP1, a mark of constitutive heterochromatin domains[12]. Here, we set to study whether 20q-TERRA KO clones also showed altered abundance of HP1 at telomeres. Interestingly, CHIP analysis showed significantly decreased HP1 abundance at the telomeres of 20q-TERRA KO clones (#A7, #E1, #E6, and #C4) compared to the WT clones (#1 and #2), demonstrating that TERRAs are required for HP1 deposition at telomeres (Fig. 4e).

**PRC2 complex is necessary for HP1 recruitment at telomeres.** Since we found that TERRAs are required for H3K9me3, H3K27me3, PRC2, and HP1 recruitment at telomeres, we next set up experiments to understand mechanistically the chain of events for which TERRAs are responsible. PRC2 and H3K27me3 have been previously to cooperate with H3K9me3 tri-methylation to maintain HP1 at chromatin[33]. To assess whether this is also happening at telomeres, we first generated cells with decreased EZH2 and SUZ12 levels by using two alternative methods. In particular, we downregulated these PRC2 components using shRNAs against EZH2 or SUZ12, as well as by using the specific EZH2 chemical inhibitor EPZ-6438 (Fig. 5a,b). The downregulation obtained with the EZH2 shRNAs was ~70–90%. the shRNA#2 being the most effective (Fig. 5a). The downregulation obtained with the SUZ12 shRNA was 85% (Fig. 5a). In the case of chemical inhibitors, we observed that the best EZH2 inhibition was achieved at concentration of 1 μM of the inhibitor with a 90% decrease in H3K27me3 global levels as determined by western blot (Fig. 5b). Next, we studied the changes at telomeres as a consequence of downregulating the PRC2 components. As expected, we found a significant decrease in global SUZ12 levels as detected by immunofluorescence in cells treated with either the EZH2 or SUZ12 shRNAs or with the EZH2 inhibitor (Fig. 5c, left bottom graphs). Consequently, the colocalization of SUZ12 with the TRF2 telomeric protein decreased significantly both with the shRNAs or upon chemical inhibition (Fig. 5c, right bottom graphs), confirming the interaction of SUZ12 with the telomere. Next, we evaluated the consequences of reduced PRC2 levels in the deposition of heterochromatic marks by telomeric ChIP. As expected because of the interaction of SUZ12 with the telomere, telomeric H3K27me3 abundance decreased significantly in cells with reduced EZH2 or SUZ12 levels (Fig. 5d, left panel). Moreover, the levels of H3K9me3 and H4K20me3 also significantly dropped in these cells (Fig. 5d, middle and right panels), indicating that PRC2 is not only important for the methylation of H3K27 but also for the deposition of these other heterochromatic marks. As a consequence of the drop in heterochromatic marks, both global HP1 levels (Fig. 6a, left bottom graphs) and its colocalization at telomeres (RAP1-HP1 colocalizations; Fig. 6a, right bottom graphs) significantly decreased both in cells treated with either the PRC2 shRNAs or upon EZH2 inhibition as observed by double immunofluorescence of HP1 and the telomeric protein RAP1. Although the presence of HP1 is a well-established mark at the telomere[12,38], we performed an additional control to prove the specificity of the HP1 antibody and of this interaction consisting of a double immunofluorescence of HP1 and the telomere-binding protein RAP1 run in parallel with one

**Fig. 5** The PRC2 complex is critical for the deposition of heterochromatic histone marks and HP1 at telomeres. **a** Upon infection of U2OS cells with shRNAs against EZH2 and SUZ12, total protein was obtained and used for western blot detection of EZH2 and SUZ12. Actin was used as loading control. **b** U2OS were treated with increasing concentrations of the EZH2 inhibitor EPZ-6438 for 4 days. Nuclear protein extracts were used for western blot detection of H3K27me3. Actin was used as loading control. (Graph) Quantification is shown. **c** Representative images of the average number of colocalizations for TRF2 (green) and SUZ12 (red) in U2OS cells infected with a scramble or with EZH2 or SUZ12 shRNAs (left panel) or treated with vehicle or EZH2 inhibitor. Arrowheads indicate colocalization events. Scale bar, 10 μm. Below the images is shown the quantification of (left graphs) the total nuclear SUZ12 upon shRNAs or EZH2 inhibitor and (right graphs) the colocalization between TRF2 and SUZ12 (mean values ± s.e.m., n = number of cells). **d** Telomeric ChIP-dot-blot of the H3K27m3, H3K9m3, and H4K20m3 for U2OS cells infected with scramble, EZH2, or SUZ12 shRNAs. IgG was used as a control. IgG ChIP-dot-blot shared by different antibodies shows different exposure times according to the best exposure time required for each antibody. DNA input signal is also shown. Quantification of the immunoprecipitated telomeric repeat signal normalized by the input for each individual sample is shown below (mean values ± s.e.m., n = technical replicates). Student's t-test was used for the statistical analysis (*$p < 0.05$, **$p < 0.01$, and ***$p < 0.001$)

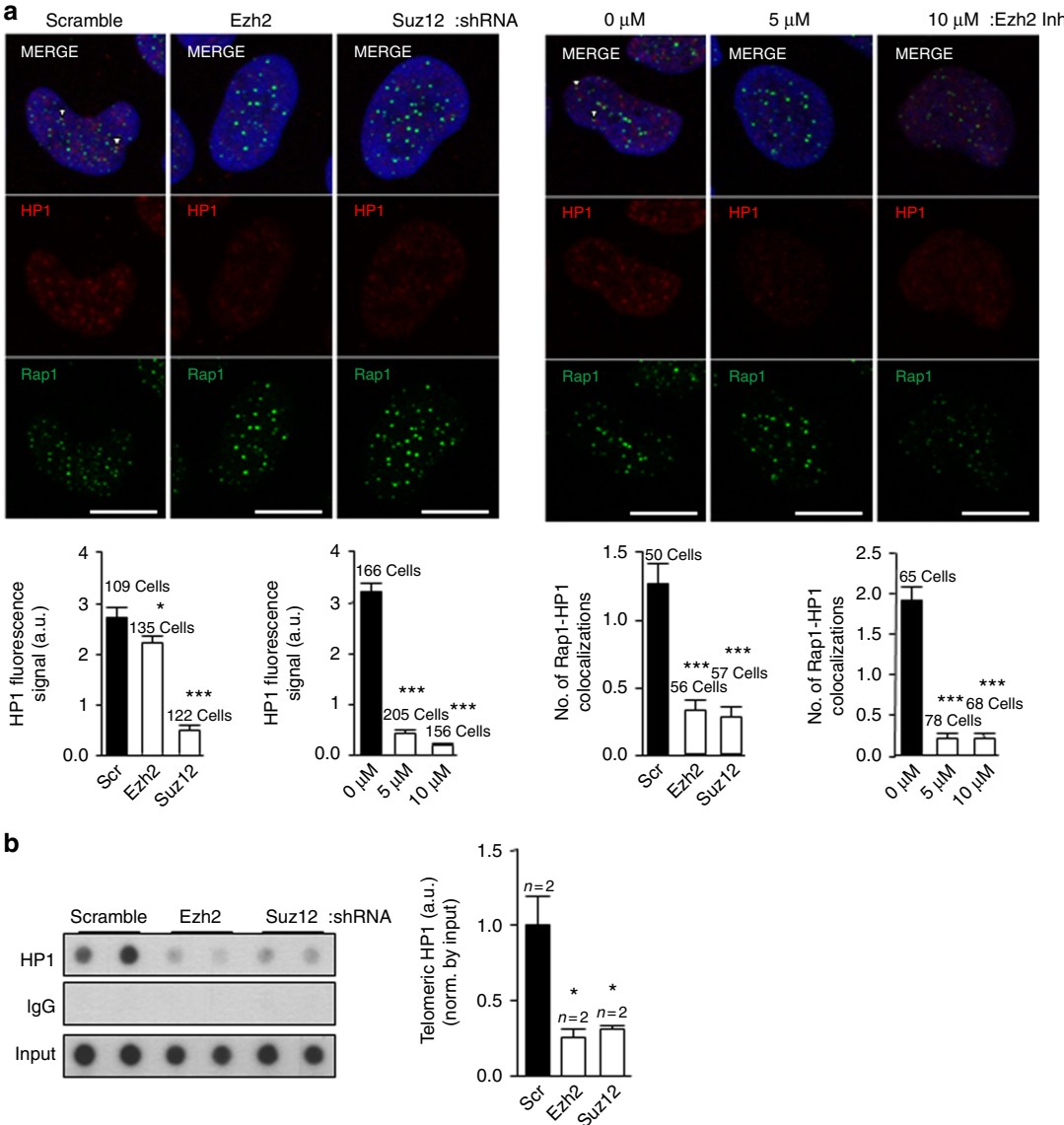

**Fig. 6** Global HP1 and its localization at telomeres depend on PRC2 levels. **a** Representative images of the average number of colocalizations of RAP1 (green) and HP1 (red) in the U2OS infected with scramble, EZH2, or SUZ12 shRNA (left panel) or in U2OS treated with vehicle or with EZH2 inhibitor. Arrowheads indicate colocalization events. Scale bar, 10 μm. Quantification of (left graphs) total HP1 and (right graphs) the colocalization between RAP1 and HP1 (mean values ± s.e.m., $n =$ number of cells). **b** Telomeric ChIP-dot-blot of the HP1 protein for the U2OS cells infected with scramble, EZH2, or SUZ12 shRNA. IgG was used as a control. DNA input signal is also shown. Quantification of the immunoprecipitated telomeric repeats normalized by the input is shown below (mean values ± s.e.m., $n =$ technical replicates). Student's $t$-test was used for the statistical analysis (*$p < 0.05$, **$p < 0.01$, and ***$p < 0.001$)

using ACA (antibody that recognizes the centromere) and RAP1. As can be seen in Supplementary Fig. 4, the number of colocalizations between HP1 and RAP1 was significantly higher than the ones observed by chance between RAP1 and the ACA signal. Similar to SUZ12 interaction with the telomere (see above), we performed two additional randomization approaches to prove the HP1–RAP1 interaction. First, using an interaction plugin on Fiji[37] we found that the strength of the interaction HP1–RAP1 is superior to zero, indicating that the spatial distribution of RAP1-HP1 is dependent (Supplementary Fig. 5). When compared against 10,000 Monte Carlo samples of NND distributions corresponding to the null hypothesis of "no interaction", the results are statistically significant ($p < 0.0001$). Second, using the randomization tool of the Definiens Developer XD.2 software, we first detected digitally RAP1 spots in the original and in 4.800

randomized pictures. (Supplementary Fig. 6A). The number of RAP1 spots found in four original pictures and in the ones randomized is shown in the Supplementary Fig. 6B (please, note that each picture contains different number of nuclei). Next, the colocalizations between RAP1 and HP1 were identified both in the original and in the randomized pictures (see Methods). Importantly, we found that the number of random colocalizations was significantly lower than in the original pictures (Supplementary Fig. 6C). The percentage of randomized pictures with ≥colocalizations is significantly lower than in the original pictures ($p > 0.01$; Supplementary Fig. 6D). Moreover, we confirmed the significant decrease in HP1 abundance at telomeres by telomeric ChIP in cells with reduced EZH2 or SUZ12 levels (Fig. 6b). Note that HP1 levels do not change upon downregulation of either EZH2 or SUZ12 (Supplementary Fig. 7)

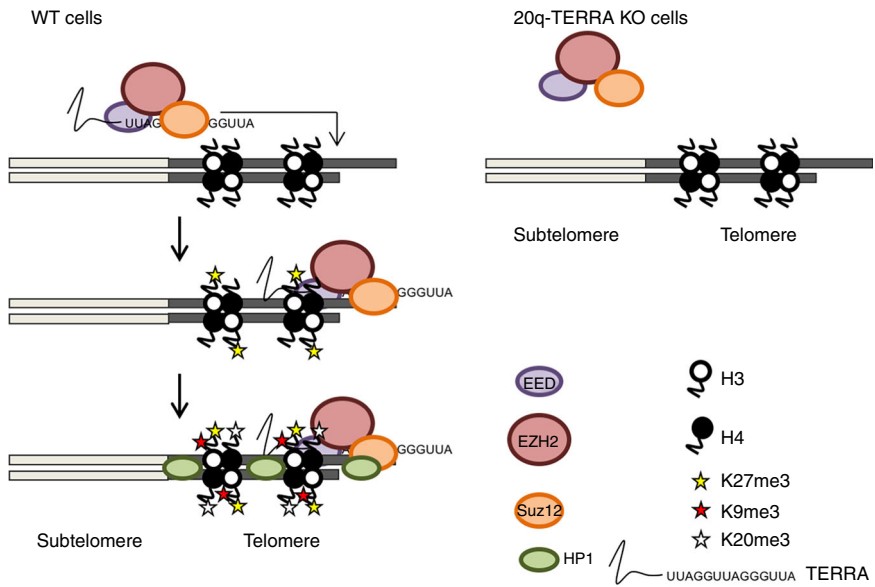

**Fig. 7** Model of TERRAs as a master regulator of the heterochromatic status of the telomere. Diagram showing how TERRA recruits the PRC2 complex (EED, EZH2, SUZ12) and directs it to the telomere. Upon binding of PRC2 to the telomere, PRC2 catalyzes H3K27 methylation. This mark facilitates then the deposition of H3K9m3 and H4K20m3 and the recruitment and stabilization of HP1 protein at the telomere, important to maintain the heterochromatic status of the telomere. In the 20q-TERRA KO cells PRC2 is not recruited to the telomere, and the heterochomatic marks and HP1 protein are not taking place

Together, these results are in agreement with a model in which TERRAs interact with the PRC2 complex, and this interaction is important for PRC2 deposition to telomeres. In turn, this deposition is critical for the establishment of the H3K27me3 histone mark at telomeres, as well as for the subsequent assembly of H3K9me3 and H4K20m34 heterochromatin marks. Upon the establishment of these marks, HP1 is recruited to telomeres (Fig. 7).

**TERRAs do not affect subtelomeric and total DNA methylation**. Given the role of TERRAs in heterochromatin assembly at telomeric chromatin, we next studied whether TERRAs can also influence subtelomeric DNA methylation as well as global DNA methylation. Subtelomeric DNA is highly methylated at CpG-rich regions[10,11], being the methylation of the D4Z4 repeat representative of the methylation state of the subtelomere. For this reason, we performed bisulfite pyrosequencing to evaluate the DNA methylation levels at the D4Z4 repeats in our panel of cells WT and 20q-TERRA KO. We did not observe differences in the different CpG analyzed (cytosines C1, 2, and 3) between the 20q-TERRA KO clones and the WT clones (Supplementary Fig. 8). The slight differences found between clones of the same genotype might be related to inner differences between the clones (Supplementary Fig. 8A). When we calculated the percentage of changes in methylated CpG of the 20q-TERRA KO clones with respect to the control, we did not find any statistical differences between the WT clones and the 20q-TERRA KO clones, in any of the cytosine evaluated (Supplementary Fig. 8B).

To assess the role of TERRAs in total DNA methylation, we performed bisulfite pyrosequencing on LINE1 repeats, which are representative of the global DNA methylation state. As with the subtelomeric DNA methylation, we did not find differences between the WT and the 20q-TERRA KO clones (Supplementary Fig. 8C). When we calculated the percent increase of methylated CpG of the 20q-TERRA KO clones with respect to the WT pool, we did not find significant differences either (Supplementary Fig. 8D). CpG methylation at Alu repeats was used as control (Supplementary Fig. 8E).

The above data indicate that TERRAs do not have a clear role on subtelomeric and total DNA methylation.

**Discussion**

Telomeres are known to be enriched in heterochromatin histone tri-methylation marks, namely H3K9me3 and H4K20me3, as well as HP1 binding[12,39,40], thus being part of the so-called constitutive heterochromatin. In addition, subtelomeric DNA is hypermethylated[7,11,13]. Both histone and DNA hypermethylation marks contribute to the repressive environment at telomeres, acting as negative regulators of telomere length and telomere recombination[8–13]. Interestingly, TERRAs are known to bind both in *cis* and in *trans* to telomeric chromatin[15,16,18]; however, the roles of TERRAs at the assembly of telomeric chromatin are still vaguely defined owing to lack of TERRA knockout models, although an interaction between TERRAs and H3K9me3 and HP1 has been described before[21].

We recently showed that the majority of TERRAs in human U2OS ALT cells arise from the 20q-TERRA locus[20]. We have also generated human U2OS cells knockout for this 20q-TERRA locus, and showed that this resulted in markedly decreased global TERRA levels, as well as loss of telomere protection and telomere shortening[20]. Here, we used this TERRA knockout cellular model to address the role of TERRAs in the establishment of telomeric chromatin. We found that human U2OS cells KO for 20q-TERRA show decreased abundance of the heterochromatin H3K9me3 and H4K20m3 histone tri-methylation marks at telomeres, previously shown by us to be important for the repressive environment of telomeres[12,39,40]. In addition, we found that DNA hypermethylation of subtelomeric sequences, which is another feature of telomeric chromatin[11], was not affected by TERRA depletion, indicating that this mark is independent of TERRA levels. Therefore, our results demonstrate that TERRAs are necessary for the establishment of the bona fide H3K9me3 and H4K20m3 heterochromatic histone marks at telomeres but not for subtelomeric DNA hypermethylation.

Interestingly, we found that 20q-TERRA KO cells also showed a decreased abundance of the facultative heterochromatin mark

H3K27me3 at telomeres, which has only been described in lower eukaryotes such as unicellular algae[41]. In the filamentous fungus *Neurospora crassa* the H3K27me2/3 mark is principally sub-telomeric covering ≈7% of the genome, and its loss results in telomere mislocalization and alterations in normal chromosome conformation[42]. As this mark is established by the Polycomb Repressive Complex 2 (PRC2) we next seek whether there was a direct interaction between TERRAs and the PRC2 complex. We found that TERRAs directly bound to the PRC2 components EZH2 and SUZ12. Furthermore, we saw that SUZ12 specifically colocalized with telomeric chromatin by different techniques, including immunoprecipitation of SUZ12 followed by telomeric dot-blot and, more importantly, by colocalization of SUZ12 with the telomeric binding protein TRF2, using super-resolution confocal microscopy. Interestingly, this interaction was decreased in the absence of TERRAs. These findings are in agreement with a recent report indicating that TERRA location to extratelomeric sites influences H3K27 deposition at these sites through interaction with the PRC2[17].

Our data support a role of TERRAs as key factors in the deposition of heterochromatic marks at telomeres in ALT human cells. It was shown before that TERRA binds H3K9me3 and HP1 (ref. [21]) but here we have completed the picture with the finding of PRC2 being the TERRA mediator for the heterochromatization of human telomeres. Thus, we found that TERRAs interact with PRC2 and that PRC2 depletion leads to loss of heterochromatic marks at telomeres. Interestingly, genome-wide targeting of PRC2 depends on ATRX[43]. However, U2OS cells lack ATRX and, therefore, it is quite possible that both PRC2 localization to telomeres and the interplay between facultative and heterochromatic marks at telomeres in U2OS cells is very different from those in telomerase-positive cells. Moreover, ALT cells are defective in the cell cycle regulation of TERRA and neither the levels of TERRA nor the colocalization of TERRA at telomeres declined from S to $G_2$ as it occurs in telomerase-positive cells[22]. This persistent TERRA presence at telomeres might be important for PRC2 recruitment in the absence of ATRX. On the other hand, the binding of TERRAs to PRC2 is conserved in mammals as it was also identified to occur in mouse embryonic stem cells[17]. In this murine cells, TERRA binding was strongly correlated with H3K27me3 genome wide[17], supporting our finding of TERRA regulating H3K27me3 levels in human telomeres. We went one step further and found that TERRAs do not only regulate H3K27me3 deposition but also H3K9me3, H4K20me3 and HP1 and that this is mediated by PRC2. According to the role described for PRC2 and H3K27me3 cooperating with H3K9 methylation to maintain HP1 at chromatin[33], our findings at telomeres are also in agreement with this heterochromatization process. Moreover, our data indicate that TERRAs would be the key molecule to lead this process. Whether this important role could be extensive to telomerase-positive cells awaits to be tested. In summary, we describe for the first time a role for PRC2 in the establishment of telomeric chromatin, which is regulated by TERRAs. In addition, we demonstrate an important role for TERRAs in telomeric heterochromatin assembly.

## Methods

**Cells, transfection, infection and treatments**. Human U2OS (ATCC) were cultured according to the ATCC's recommendations. Plasmids were electroporated using the Neon Transfection System (Invitrogen) following the manufacturers' protocol. U2OS cells were infected with an shRNA against the PRC2 complex subunits, which were cloned into Plko.1-Puro vector (shRNA sequence available in Supplementary Table 1). Lentiviruses were packaged in 293T cells (ATCC-CRL-3216) using the third-generation packaging system vectors, pMDLg/pRRE, pRSV. Rev, pMDG VSVG. Cells were seeded at a 50% of confluency 24 h before infection. Two infections were performed every 24 h by adding 3 ml of viral supernatant. Then, cells were allowed to recover for 24 h in growth medium before undergoing

selection with puromycin for 2 days. For EZH2 inhibition U2OS cells were cultured with DMEM that contained different concentrations of the EZH2 inhibitor EPZ-6438 (SELLECKCHEM) for 4 days.

**Generation of TERRA KO clones using the CRISPR-Cas9 system**. The 20q-TERRA KO clones were generated using an all-in-one plasmid that contains the two gRNAs (S2 and E1) needed for the deletion of the 20q-TERRA locus. The backbone of the all-in-one plasmid was the pSpCas9(BB)-2A-GFP that already contains the S2 gRNA[20]. The gRNA End1 (E1) with the U6 promoter was amplified by PCR adding Not1 restriction sequence (primers available in Supplementary Table 1) with the GoTaq polymerase (Promega) using as template the pSpCas9(BB)-2A-GFP that already contains the E1 gRNA[20]. The PCR product and the pSpCas9(BB)-2A-GFP containing the Start 2 (S2) gRNA[20] were digested with the Not1 enzyme (New England Biolabs) during 4 h at 37 °C, and after ligation was performed at 16 °C overnight with the T4 DNA ligase enzyme (New England Biolabs). The plasmid with the new insert was transformed in One Shot TOP10 Chemically Competent E. coli DH5α (Invitrogen) and plated on LB agar plates with penicillin. The plasmid was purified with Plasmid Miniprep Kit (QIAGEN) and the presence of the insert confirmed with Sanger-style BigDye terminator chemistry on an ABI 3730 × l sequencer (Applied Biosystems). Then, the all-in-one plasmid containing the Cas9, both gRNA (S2 and E1), and a GFP cassette was transiently electroporated into the cells using the Neon system. Two days after transfection cells were trypsinized, washed with dPBS, and three or five cells of the 10% GFP brightest ones were sorted using the FACS ARIA IIU (Becton Dickinson) and plated into 96-well plates. Two weeks later, wells were checked for monoclonal cell expansion and those clones were selected for genotyping by PCR. Homozygous monoclonal cell lines were expanded. PCR primers used for genotyping can be found in Supplementary Table 1.

**RNA dot-blot**. RNA dot-blot analyses were performed using standard protocols. TERRA probe was obtained from a 1.6-kb $(TTAGGG)_n$ cDNA insert excised from pNYH3 (kind gift from T. de Lange, Rockefeller University, NY, USA). Dot-blot was normalized using 18S probes and quantified using ImageJ. The probes were labeled using the commercial Prime-It II Random Primer Labeling Kit (Agilent Genomics).

**RNA-FISH**. Cells grown on poly-L-lysine-coated coverslips (Becton Dickinson) were placed in cytobuffer (100 mM NaCl, 300 mM sucrose, 3 mM $MgCl_2$, 10 mM pipes pH 6.8) for 30 s, washed in cytobuffer with 0.5% Triton X-100 for 30 s, washed in cytobuffer for 30 s, and then fixed for 10 min in 4% paraformaldehyde in phosphate-buffered saline (PBS). The cells were dehydrated in 70, 80, 95, and 100% ethanol, air-dried, and hybridized overnight at 37 °C with a telomere-specific PNA-FITC probe (Panagene) in hybridization buffer (2 × sodium saline citrate (SSC)/50% formamide). Coverslips were washed two times for 15 min in hybridization buffer at 40 °C, two times for 10 min in 2 × SSC at 40 °C, 10 min in 1 × SSC at 40 °C, 5 min in 4 × SSC at room temperature, 5 min in 4 × SSC containing 0.1% Tween-20 and DAPI (Molecular Probes) at room temperature, and 5 min in 4 × SSC at room temperature. Signals were visualized in a confocal ultraspectral microscope SP5-WLL (Leica). TERRA signal was quantified using the Definiens Developer XD.2 software.

**High-throughput telomere length quantification by FISH**. HT-q-FISH was done according to ref. [44]. Cells were plated into clear-bottom black-walled 96-well plates pre-coated for 30 min with 0.1% porcine gelatin. Then, cells were grown in DMEM at 37 °C overnight, fixed in methanol/acetic acid (3:1, *v/v*) two times for 10 min, and were left in the fixative overnight at −20 °C. Fixative was removed, plates dried for at least 1 h at 37 °C, and samples were rehydrated in PBS. Plates were then subjected to a standard Q-FISH protocol using a telomere-specific PNA-CY3 probe; DAPI was used to stain nuclei. Images were captured using the OPERA (Perkin Elmer) High-Content Screening system. TL values were analyzed using individual telomere spots. Samples were analyzed in triplicate.

**Biotin pull-down analysis**. Biotin pull-down assays were carried out as described in ref. [45] except for that 150 μg of nuclear lysate was incubated with 0.9 ng of biotinylated transcripts (Invitrogen) for 1 h at room temperature. An amount of 0.9 ng of total RNA was added as competitor. Complexes were isolated using streptavidin-conjugated Dynabeads (Dynal), and bound proteins in the pull-down material were analyzed by western blotting. The biotinylated TERRA sense transcript consists of 8 × UUAGGG; the control biotinylated transcripts consist of the TERRA antisense sequence 8 × CCCUAAA. See antibodies in the western blot section. Full gels can be found in Supplementary Fig. 9.

**Western blot analysis**. Whole-cell lysates and nuclear lysates were prepared using RIPA buffer and RSB buffer (10 mM Tris-HCl (pH 7.5), 10 mM NaCl, 3 mM $MgCl_2$, and inhibitors), respectively, as previously described[19]. Protein lysates were resolved by SDS–PAGE and transferred onto nitrocellulose membranes. Antibodies used were the following: EZH2 (D2C9; Cell signaling), SUZ12 (ab12073; Abcam), H3K27m3 (07–449; Millipore), HP1γ (clone 42S2; Millipore), and ACTIN (A5441;

Sigma-Aldrich). Following secondary antibody incubations, signals were visualized by enhanced chemiluminescence. Full gels can be found in Supplementary Fig. 9.

**Immunofluorescence and super-resolution microscopy.** Cells grown on poly-L-lysine-coated coverslips (Becton Dickinson) were placed in cytobuffer (100 mM NaCl, 300 mM sucrose, 3 mM MgCl$_2$, 10 mM pipes pH 6.8) with 0.5% Triton X-100 for 6 min and then fixed for 10 min in 4% paraformaldehyde in PBS. Cells were blocked with 10% BSA in PBS for 1 h at room temperature. Cells were incubated with primary antibody dissolved in Dako antibody Diluent (Dako) for 1 h in a humid chamber at room temperature. Coverslips were washed three times for 30 min in PBS containing 0.1% Tween-20. Cells were incubated with Alexa secondary antibody (Life Technologies, A11017) dissolved in Dako antibody Diluent (Dako) for 1 h in a humid chamber at room temperature. Cells were washed three times for 30 min in PBS containing 0.1% Tween-20. Samples were mounted in Prolong with DAPI (Invitrogen). Signals were visualized in confocal ultraspectral microscope SP5-WLL (Leica). The following antibodies were used: TRF2 (clone 4A794; Millipore) diluted 1/250, SUZ12 (ab12073; Abcam) diluted 1/500, RAP1 (A300–306A; Bethy) diluted 1/250, HP1γ (clone 42S2; Millipore) diluted 1/200, and Anti Centromere Antibody (ACA; 15–235; Antibodies Incorporated) diluted 1/100. For Super-resolution microscopy acquisition, we use a confocal multispectral Leica TCS SP8 system with a 3 × STED module for super-resolution. Laser lines: 405 nm and WLL2 (white laser for 470–670 nm excitation). Depletion lines: 592 and 660 nm.

**Randomization analysis.** Two different randomization analyses were performed. First, the MosaicIA Fiji plugin (https://imagej.net/Interaction_Analysis)[37,46] was used to calculate the interaction potentials. Potential used was Plummer (100 iterations). Grid spacing was 0.5 pixels. Kernel wt(q) was 0.001. Kernel wt(p) was provided by the plugin using Silverman's rule. The results were tested for significance against 10,000 Monte Carlo samples of point distributions corresponding to the null hypothesis of "no interaction"[47]. Second, the randomization tool of the Definiens Developer XD.2 software was used. TRF2 spots were found in DNA-positive nuclear areas. Those regions without DNA signal, where TRF2 is not found but SUZ12 is detected (as in the nucleoli), were eliminated from the analysis. Colocalizations between TRF2 and SUZ12 were identified according to SUZ12 signal intensity and a colocalization of TRF2 signal on SUZ12 of more than 60%. The number of colocalizations between SUZ12 and the virtual TRF2 spots were calculated using the same criteria for the original and randomized images. In all, 16,000 randomized pictures were generated. In a similar manner, the colocalizations of RAP1-HP1 were calculated in the original pictures and in 4.800 randomized pictures. The RAP1 spots were detected in the entire nucleus.

**ChIP assay and telomere dot-blots.** For ChIP analysis, we used 4 × 10^6 cells per condition. Formaldehyde was added directly to tissue culture medium to a final concentration of 1% and left for 10 min at room temperature on a shaking platform. The crosslinking was stopped by adding glycine to a final concentration of 0.125 M. Crosslinked cells were washed twice with cold PBS, scraped and lysed at a density of 5 × 10^6 cells ml^−1 for 10 min at 4°C in 1% SDS, 50 mM Tris-HCl (pH 8.0) and 10 mM EDTA containing protease inhibitors. Lysates were sonicated to obtain chromatin fragments <1 kb and centrifuged for 15 min in a microfuge at room temperature. An aliquot of 200 μl of lysate was diluted 1:10 with 1.1% Triton X-100, 2 mM EDTA, 150 mM NaCl, and 20 mM Tris-HCl (pH 8.0) containing protease inhibitors, and precleared with 50 μl of Protein A/G Plus-Agarose sc-2003 beads (Santa Cruz Biotechnology). After centrifugation, the fragments were incubated with 4 μg of the following antibodies: H3K27m3 (07–449, Millipore), H3K9m3 (07–442, Upstate), H4K20m3 (07–749, Upstate), H3K4m3 (CS200580, Millipore), H3K9ac (072K4824, Sigma-Aldrich), H4K16ac (cat39167, Active Motif), or normal rabbit IgG (Santa Cruz Biotechnology) and 10 μg of: HP1γ (clone 42S2; Millipore) and SUZ12 (ab12073; Abcam) diluted at 4 °C overnight on a rotating platform. Next day, 50 μl of Protein A/G Plus-Agarose (sc-2003) beads were added and incubated for 1 h. The immunoprecipitated pellets were washed once with 0.1% SDS, 1% Triton X-100, 2 mM EDTA, 20 mM Tris-HCl (pH 8.0), and 150 mM NaCl, and then with 0.1% SDS, 1% Triton X-100, 2 mM EDTA, 20 mM Tris-HCl (pH 8.0), and 500 mM NaCl, and next with 0.25 M LiCl, 1% Nonidet P-40, 1% sodium deoxycholate, 1 mM EDTA, and 10 mM Tris-HCl, pH 8.0 and finally with 10 mM Tris-HCl (pH 8.0) and 1 mM EDTA two times. The chromatin was eluted from the beads twice by incubation with 250 μl 1% SDS and 0.1 M NaHCO$_3$ during 15 min at room temperature with rotation. After adding 20 μl of 5 M NaCl, the crosslink was reversed overnight at 65 °C. Samples were supplemented with 20 μl of 1 M Tris-HCl (pH 6.5), 10 μl of 0.5 M EDTA, 20 μg of RNase A, and 40 μg of proteinase K, and were incubated for 1 h at 45 °C. DNA was recovered by phenol–chloroform extraction and ethanol precipitation, slot-blotted it onto a Hybond N+ membrane, and hybridized it with a plasmid containing 1.6 Kb of TTAGGG repeats (gift from T. de Lange, Rockefeller University, USA). The signal was quantified with the ImageJ software. For total telomeric DNA samples, 500 μl of lysate was processed with the rest of the samples at the step of reversing the crosslinks. The amount of telomeric DNA immunoprecipitated was calculated in each ChIP based on the signal relative to the corresponding total telomeric DNA signal. Full blots can be found in Supplementary Fig. 9.

**Data availability**. All relevant data are available from the authors.

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

## Acknowledgements

We thank to R. Torres and S. Rodriguez for advice in the CRISPR-Cas9 technology and helpful discussions. We are indebted to Sylvia Gutiérrez Erlandsson, principal investigator of the confocal microscopy Group at the National Centre of Biotechnology, Madrid (Spain), because, without her generous help, we could not have performed the super-resolution microscopy. Research in the Blasco lab is funded by the Spanish Ministry of Economy and Competitiveness Projects (SAF2013-45111-R and SAF2015-72455-EXP), the World Cancer Research (WCR) Project (16-1177), and the Fundación Botín (Spain).

## Author contribution

M.A.B. conceived the original idea. J.J.M. conducted and analyzed the experiments. J.J.M., I.L.S., and M.A.B. designed the experiments, D.M. did the super-resolution microscopy acquisition and performed the immunofluorescence image quantification and the Definiens randomization. A.C.G. did the MosaicIA randomization analysis. M.F.F. performed the DNA methylation analysis; J.J.M., I.L.S., and M.A.B. wrote the manuscript.

## Additional information

**Competing interests:** The authors declare no competing interests.

