## [Peer Review File · Nature Communications]

Reviewers' Comments:

Reviewer #1:

Remarks to the Author:

Montero and coworkers have studied the influence of TERRA RNAs on the organization of chromatin around the telomeres. They ablate the expression of most of TERRA RNAs via deletion of a region in chromosome 20 and use these cells to show that reducing TERRA RNAs diminishes H3K9me3 and H4K20me3 tri-methylation histone marks in telomeric chromatin. TERRA RNAs also appear important for tri-methylation at H3K27m3, which is carried out by the PRC2. The authors present evidence suggesting that TERRA transcripts directly bind Ezh2 and Suz12 and thereby recruit PRC2 to telomeres. Further data suggest that PRC2 at telomeres is needed for the presence of H3K27me3, H3K9me3, and H4K20me3, and HP1 at telomeres.

The study is interesting in that it connects TERRAs with chromatin metabolism at telomeres. However, some of the authors' observations do not fully support their model and must be strengthened as indicated below.

Major comments

1) Supplemental Figures S1 and S2 appear important for the main story. Can the authors show representative data in the main article? If they do not want to have more figures, perhaps after modifying the current Fig. 1 as suggested in point 2) they can include some of these data.

2) In Figure 2, a key piece of data needs to be validated with more definitive analysis. The fact that biotinylated oligomer (8×UUAGGG) brings down Ezh2 and Suz12, as shown later on in the schematic, under the conditions tested cannot unequivocally be interpreted to mean that TERRA binds these two proteins. This pull down analysis could instead indicate that bringing down TERRA RNAs could at the same time bring down a host of complexes among which Ezh2 and Suz12 are present but do not directly bind TERRA transcripts. To distinguish between these two possibilities the authors must test these interactions vivo using more stringent methods that involve crosslinking and immunoprecipitation of Ezh2 (and separately Suz12) to see if these proteins associate with TERRA transcripts. Moreover, this analysis will inform the authors about the actual TERRA transcripts involved in this interaction. They then can gain definitive evidence of the interaction in vitro using recombinant purified TERRA RNA, Ezh2, and Suz12. These data are needed to support the authors' model.

3) In Figure 2B, the quantification and description of the data ['approximately 1.5 Suz12 spots (in red) per nucleus colocalized with TRF2 (in green)'] are not very intuitive. Out of how many TRF2 signals does Suz12 show colocalization? Does this mean that TRF2 colocalizes with Suz12 in only a few telomeres, and that most telomeres do not have Suz12?

4) In Figure 3, the recruitment by PRC2 of H3K27me3, H3K9me3, and H4K20me3, and HP1 at telomeres seems poorly supported by the immunofluorescence data, which is notoriously artifact-prone and subjective. Can the authors substantiate this evidence by using molecular biology approaches that can provide more quantifiable results?

5) The text reads well, but some minor corrections are needed.

Minor

1) The authors are encouraged to use consistently TERRA or TERRAs; the latter name seems more accurate, given that the word refers to a collective of TERRA transcripts.

2) In Figure 1, the same strips at different intensities were used for 'IgG' and 'Input' signals in all 6 panels, was this intended? If it was, the authors should mention this in the legend, as the reader

may be confused. Perhaps better, they should display a single stack of strips (all sharing single strips of IgG and input) and add letter numbers to just the graphs. The same layout could be employed in Figure 3 for a similar experiment with shared Input image strips.

Reviewer #2:

Remarks to the Author:

In their manuscript entitled 'TERRA-dependent recruitment of polycomb to telomeres is essential for the assembly of histone trimethylation marks at telomeric heterochromatin' Montero et al. investigate the involvement of the non-coding RNA TERRA in the recruitment of the Polycomb Repressive Complex 2 (PRC2) and its role in the deposition of the heterochromatic histone modifications H3K9me3, H3K27me3, and H4K20me3 at telomeres. This manuscript is an extension of their previous study 'Telomeric RNAs are essential to maintain telomeres' (Montero et al., 2016, Nature Communications 7, 12534) in which the authors identified a subtelomeric region of chromosome 20 (20q) as the source of TERRA RNA in humans. In the present study the authors use a newly derived panel of 20q-deletion cell lines in U2OS cells generated via the CRISPR/Cas9 system. Using these deletion cell lines they find that the 20q-TERRA RNA is required for deposition of the heterochromatic histone marks H3K9me3, H3K27me3 and H4K20me3 at telomeres but not for hypermethylation of subtelomeric DNA. They further show that the Polycomb Repressive Complex 2 (PRC2) binds TERRA RNA and shows localisation at telomeres and that loss of PRC2 leads to a defect in telomeric heterochromatin formation.

I am not a specialist in TERRA or telomere biology and I also have no invested interest in the topic, I can therefore provide an unbiased view. I also start out reviewing scientific manuscripts with a positive attitude and I try to provide constructive feedback. Unfortunately, the work presented in this manuscript has several fundamental flaws and does not meet basic scientific standards. As such I feel that this manuscript is not suitable for publication in Nature Communications or indeed any other scientific journal. I will explain the critical points for this conclusion below:

> The U2OS 20q-TERRA deletion cells that the authors are using in Figure 1 and Figure 2B and 2C are problematic as they are not a genetically 'clean' system. These cell lines were generated using the CRISPR/Cas9 system with 2 guide RNAs that produce a deletion of the subtelomeric 20q locus. While the deletion of the 20q locus in the new panel of clones that the authors use seems to be OK and validated by PCR (Supplementary Figure S1B) there is still substantial expression of TERRA RNA (Supplementary Figure S1C) in all clones, so clearly this is not a complete TERRA knock out and there are sources of TERRA RNA outside of the 20q locus. The expression of TERRA RNA is very heterogeneous in these cell lines (see Supplementary Figure S1C) which is also reflected in the results, e.g. the authors claim that there is no regulation of telomeric H3K4me3 modification levels while clearly the H3K4me3 levels are ranging from no reduction (clone A7) to only 40% expression compared to wild-type in 50% of the clones (E6 and C4). The same is true for H3K27me3 modification levels, while the authors argue that H3K27me3 levels are generally down-regulated clearly the H3K27me3 levels are only down-regulated in 50% of the clones (clones A7 and E1) but not at all in the other 50% (clones E6 and C4). Such levels of heterogeneity make it very difficult to draw any conclusions and all data based on these 20q-TERRA KO cells have to be treated with extreme caution.

> The main, and principle, problem with the 20q-TERRA 'knock out' cells that the authors have generated is that there is the potential for off-target effects and that these cell lines might contain additional mutations that the authors do not know about and that might cause or contribute to the observed effects. These cells were derived in a similar way as described in the original Montero et al. (2016) study. In this study the authors argue that off-target effects do not constitute a problem since they did not find mutations in 3 of the top predicted off-target loci. However, I do not follow this logic and I think that off-target mutations are a major concern in this case and completely

invalidate all experiments using these 20q-TERRA KO cells. Finding no off-target effects in a handful of bioinformatically predicted loci is not an argument for not having off-target mutations elsewhere in the genome where the authors did not check, and therefore off-target mutations cannot be excluded. This is especially problematic in this case where the sole selection criteria for the cells is long-term survival. Telomeres are essential for long term-cell viability and if indeed the 20q-deletion is a TERRA knock out the clones that survive will exactly be those that have gained additional mutations that will help to bypass the problem and allow the cells to survive. The very fact that these cells are viable makes it very likely that the clones that the authors have generated contain additional mutations that compensate for the 20q-deletion and that are somehow involved in telomere regulation processes. The clones they have isolated therefore have to be regarded as compromised and the results obtained are not trustworthy.

In the original Montero et al. (2016) study the authors tried to generate 20q-deletion cells from 4 different cell lines (IMR90, HeLa, HCT116 and U2OS). None of the homozygous IMR90, HeLa, and HCT116 20q-deletion clones survived, the only cell line that produced viable clones were U2OS cells and the survival rate of the homozygous U2OS clones was also only 50%. The authors argued that this is due to higher TERRA expression levels in U2OS cells. On the one hand this demonstrates that the 20q-TERRA locus is essential and that the authors are onto something. On the other hand this creates major problems for the present manuscript. Firstly, U2OS cells are obviously outliers and therefore in my opinion not a suitable model to study telomere biology in general. Secondly, upon observing the lethality induced by deleting the 20q-TERRA locus the authors should not have carried out further experiments with these 'promiscuous' U2OS cell lines where results cannot be trusted. Instead they should have established a conditional 20q-TERRA knock out model in a different cell line in which the locus can be inducibly deleted and in which it can be excluded that the cells acquire additional mutations that could influence the results. In summary and as stated above, any data obtained from the U2OS 20q-TERRA KO cells in this study are invalid as it is not clear whether the observed effects are due to the 20q-TERRA deletion or another unknown mutation. The data presented in the entire Figure 1 and in Figures 2B and 2C can therefore not be used to make the claim that the 20q-TERRA RNA is involved in establishing heterochromatin at telomeres.

> In addition, the immunofluorescence assay to demonstrate telomeric localisation of Suz12 and HP1 is entirely inadequate. Just from looking at the immunofluorescence images it is clear that the stainings for telomere factors (TRF2 in Figures 2B and 3C and Rap1 in Figure 3F) and heterochromatin factors (Suz12 in Figures 2B and 3C and HP1 in Figure 3F) are clearly different and that there is no distinctive localisation of Suz12 and HP1 to telomeres. The number of Suz12 and HP1 fluorescence signals (1.5 for Suz12 and 3 for HP1) per cell cannot be quantified, as I assume the authors have done, like a dotted pattern since clearly both Suz12 and HP1 show diffuse nuclear stainings in the wild-type cells with some regions staining more intensely, as expected. The very low number of co-localisations that the authors report based on their quantifications are likely to result from just chance overlaps of fluorescence signals between the multi-punctuate telomere staining patterns of TRF2 and Rap1 within the diffuse Suz12 and HP1 stainings. The spatial resolution of a normal confocal microscope, such as the Leica SP5-WLL used here, does not allow one to resolve co-localisations at a level that the authors require. The authors can therefore not claim a localisation of Suz12 or HP1 to telomeres, unless they can unambiguously demonstrate and quantify this with approaches that allow identification of bona fide co-localisations at high resolution and with 3D reconstructions. As such the experimental approach presented in Figures 2B and 3C and in Figure 3F is inadequate, and therefore the data are not valid to make the claim that the 20q-TERRA RNA and PRC2 are required for the recruitment of heterochromatin factors to telomeres. Even if the co-localisations were real, around one TRF2-Suz12 (Figures 2B and 3C) or Rap1-HP1 (Figure 3F) co-localisation event over around 50 telomere spots per cell in wild-type cells, as quantified by the authors, does not indicate extensive telomeric localisation and argues against this phenomenon being a general mechanism.

The authors might have a point in demonstrating that the PRC2 complex is involved in establishing telomeric heterochromatin since depletion of the subunits Ezh2 and Suz12 by siRNA leads to loss

of HP1, H3K9me3 and H4K20me3 at telomeres, but this has not been investigated further in any detail.

Given the low quality of most of the presented data and that large parts of the manuscript are invalid due to flawed scientific approaches, there is not much valid data that can be taken into account to support the findings claimed by the authors and therefore this manuscript should be rejected.

Reviewer #3:

Remarks to the Author:

In this study, Montero and colleagues established additional U2OS clones/lines KO for the 20q-TERRA locus. This is important, given that U2OS cells do not maintain their telomere length within a narrow range and may exhibit large inter-clonal variation. Using these and previously established clones the authors now show that TERRA is partly responsible for the establishment of the H3K9me3 and H4K20me3 constitutive heterochromatin marks (previously known to present at telomeres) as well as a facultative heterochromatin mark H3K27me3 (not known to be present) at telomeres. They further show that 8×UUAGGG oligos can interact in vitro with the PRC2 complex subunits Ezh2 and Suz12, which are responsible for H3K27 tri-methylation, and that rare instances of PRC2 and TRF2 co-localization are TERRA-dependent. Finally, they provide evidence that PRC2 not only deposits H3K27me3, but also plays a role in accumulation of both H3K9me3 and HP1 at telomeres.

These findings extend our knowledge of the telomeric heterochromatin composition and the way it is established. However, there are a number of caveats and specific comments that must be addressed

1. Previously, Maria Blasco lab has demonstrated that the majority of TERRA in human cells is transcribed from 20q (Montero et al., Nat Comm 2016). This was confirmed by CRISPR-Cas9-mediated bi-allelic knockout of the putative 20q-TERRA locus, which also allowed to show the importance of TERRA for telomere length maintenance. Curiously, the biallelic deletion of the 20q-TERRA locus was only possible in U2OS cells that maintain their telomeres by ALT, but not in the telomerase-dependent cell lines. The reason for that remains obscure, since U2OS cells that have upregulated TERRA level should be more rather than less sensitive to TERRA KO. In any case, it's important to bear in mind that the studies of TERRA knockout are currently limited to the U2OS cell line that has a very peculiar way of telomere maintenance and therefore it remains unclear whether the findings can be generalized onto telomerase-dependent cells. This point is crucial and the authors must be thoroughly addressed.

2. In general, the authors confirmed that a large fraction of TERRA is transcribed from the putative 20q-TERRA locus, although analysis of the multiple clones now demonstrated that it's responsible for only 20-50% of human TERRA. This point should be highlighted given discrepancies between the Blasco's Lab and other telomere labs.

They also confirmed that 20q TERRA KO results in telomere length reduction: on average KO clones have shorter telomeres compared to controls. However, it's disconcerting that there is no clear correlation between the level of TERRA and telomere length reduction when individual clones are considered (Sup. Figs 1C and 2). It's a pity that the previously established clone C4 that has by far the shortest telomeres (Sup. Fig. 2) has not been analyzed for TERRA level along with the other clones (Sup. Fig. 1).

3. Similarly, on average, the 20q-TERRA KO clones have reduced H3K9me3 and H4K20me3 at telomeres (Fig. 1A,B), but there is no correlation between the levels of TERRA (Sup. Fig. 1C,D) and these constitutive heterochromatin marks (Fig. 1A,B). In fact, clone E6 that has the lowest TERRA level (Sup. Fig. 1C) shows the levels of H3K9me3 and H4K20me3 that are the least

different from the controls. This lack of “dose-response” relationship between the levels of TERRA and constitutive heterochromatin marks suggests a complex and indirect relationship between the two. The same comment goes for the facultative heterochromatin mark, H3K27me3. How the authors can reconcile this lack of correlation

4. The interaction between the biotinylated 8×UUAGGG oligos and the Ezh2 and Suz12 subunits of PRC2 in cell lysates (Fig. 2A) is convincing, but this interaction has already been demonstrated *in vivo* in mESCs using iDRIP-Mass Spec (Chu et al., Cell 2017), which needs to be acknowledged. A useful control here would be a non-telomeric G-rich oligo to determine whether PRC2 has a general affinity for G-rich RNA especially those capable to fold into G-quadruplex. The effect of 20q-TERRA KO on co-localization between Suz12 and TRF2 (Fig 2B) is less convincing, mainly due to the rarity of these co-localization events (1-2 per nucleus). The number of nuclei analyzed for each clone has to be indicated. This experiment should be strengthened.

5. The decrease of the H3K27me3 mark at telomeres in response to either Ezh2 or Suz12 knockdown analyzed by CHIP is very minor, whereas decrease of the H3K9me3 and especially of the H4K20me3 is much more pronounced (Fig 3D). This does not make sense. The IgG background in H3K27me3 CHIP is very high, which perhaps is a problem. To make the data more convincing, it would be nice to repeat the latter. Instead of “recruitment” (line 249) use “deposition” of heterochromatic marks.

6. Reduction of the HP1 at telomeres in response to either PRC2 knockdown or Ezh2 inhibition analyzed by IF (Fig. 3E) is difficult/impossible to evaluate due to very poor and noisy HP1 signal, but it is quite convincing by CHIP (Fig. 3F). To make a “clean” conclusion regarding the role of PRC2 in HP1 recruitment, the authors should show that Ezh2 knockdown does not result in downregulation of HP1 protein level (by western blot).

7. This study provides an additional insight in the composition of the telomeric chromatin (in the ALT cell line), and uncovers the unexpected role of PRC2 in the accumulation of constitutive heterochromatin marks and HP1 at telomeres. How exactly PRC2 performs this function remains unclear. It's worthy to note that genome-wide targeting of PRC2 depends on ATRX (Sarma et al., Cell 2014). Remarkably, U2OS cells used in this study are ATRX-null. Therefore, it is quite possible that both PRC2 localization to telomeres and the interplay between facultative and heterochromatic marks at telomeres in U2OS cells is very different from those in telomerase-positive cells. The authors may consider mentioning this possibility in the discussion.

Point-by-point answer to the reviewers

Detailed answer to reviewer #1

[REVIEWER]: Montero and coworkers have studied the influence of TERRA RNAs on the organization of chromatin around the telomeres. They ablate the expression of most of TERRA RNAs via deletion of a region in chromosome 20 and use these cells to show that reducing TERRA RNAs diminishes H3K9me3 and H4K20me3 tri-methylation histone marks in telomeric chromatin. TERRA RNAs also appear important for tri-methylation at H3K27m3, which is carried out by the PRC2. The authors present evidence suggesting that TERRA transcripts directly bind Ezh2 and Suz12 and thereby recruit PRC2 to telomeres. Further data suggest that PRC2 at telomeres is needed for the presence of H3K27me3, H3K9me3, and H4K20me3, and HP1 at telomeres.

The study is interesting in that it connects TERRAs with chromatin metabolism at telomeres. However, some of the authors' observations do not fully support their model and must be strengthened as indicated below.

[AUTHORS]: We are grateful to this reviewer for considering that “**the study is interesting in that it connects TERRAs with chromatin metabolism at telomeres**”. In addition, we thank the reviewer for the detailed review of our manuscript and the very useful commentaries and suggestions, which we have addressed in the revised manuscript.

Major comments

[REVIEWER]: 1) Supplemental Figures S1 and S2 appear important for the main story. Can the authors show representative data in the main article? If they do not want to have more figures, perhaps after modifying the current Fig. 1 as suggested in point 2) they can include some of these data.

[AUTHORS]: As suggested by reviewer #1, we have moved the Suppl. Fig S1 and S2 to main figures (see new Fig. 1 and 2) and we have revised the text accordingly (see page 27, lane 1 to page 28, lane 7).

[REVIEWER]: 2) In Figure 2, a key piece of data needs to be validated with more definitive analysis. The fact that biotinylated oligomer (8Å-UUAGGG) brings down Ezh2 and Suz12, as shown later on in the schematic, under the conditions tested cannot unequivocally be interpreted to mean that TERRA binds these two proteins. This pull down analysis could instead indicate that bringing down TERRA RNAs could at the same time bring down a host of complexes among which Ezh2 and Suz12 are present but do not directly bind TERRA transcripts. To distinguish between these two possibilities the authors must test these interactions vivo using more stringent methods that involve crosslinking and immunoprecipitation of Ezh2 (and separately Suz12) to see if these proteins associate with TERRA transcripts. Moreover, this analysis will inform the authors about the actual TERRA transcripts involved in this interaction. They then can gain definitive evidence of the interaction in vitro using recombinant purified TERRA RNA, Ezh2, and Suz12. These data are

needed to support the authors' model.

[AUTHORS]: We agree with the reviewer's comment that the biotin pull-down assay with nuclear extracts cannot exclude the possibility of indirect interactions. We did not include any direct interaction assay in the original manuscript because this was already published during the preparation of our manuscript. In particular, Wang et al used a recombinant holo-PRC2 5-mer complex (EZH2, EED, SUZ12, RBBP4, and AEBP2) to prove the binding of the PRC2 complex to TERRA by electrophoretic mobility shift assay (EMSA) (Wang et al., Mol. Cell, 2017). Later on, Chu and coworkers proved the direct interaction between TERRA and Ezh2 by iDRIP (identification of direct RNA interacting proteins) (Chu et al., Cell 2007). We have now revised the manuscript to include these published data that reinforces our findings (**see page 10, lanes 19-23; page 11, lanes 1,2**).

[REVIEWER]: 3) In Figure 2B, the quantification and description of the data ['approximately 1.5 Suz12 spots (in red) per nucleus colocalized with TRF2 (in green)'] are not very intuitive. Out of how many TRF2 signals is does Suz12 show colocalization? Does this mean that TRF2 colocalizes with Suz12 in only a few telomeres, and that most telomeres do not have Suz12?

[AUTHORS]: Yes, in fact only few telomeres colocalize with Suz12 as determined by double immunofluorescence with Suz12 and TRF2 antibodies, in particular 3.5% spots/nuclei of TRF2 reproducibly colocalize with Suz12. Importantly, the colocalization between TRF2 and Suz12 is specific as it is significantly decreased upon down-regulation of either Suz12 or the Suz12-interactor protein Ezh2 by shRNA or upon treatment with a chemical Ezh2 inhibitor (see Fig. 5). Moreover, we also demonstrate the consequences of this interaction between PRC2 components and telomeres by measuring the activity of the PRC2 complex at telomeres. In particular, we demonstrate presence of H3K27me3 at telomeres by telomeric ChIP dot blot and show that deposition of this mark at telomeres is dependent on PRC2 levels (see **new Fig. 5**). Nevertheless, we have now performed additional experiments to strengthen this point. In particular, we have demonstrated the interaction of the PRC2 complex with the telomere by performing a Suz12 ChIP followed by telomeric dot-blot to detect telomeric DNA (see "scramble shRNA" in **new Fig. 4C**). Furthermore, we show that this interaction is significantly decreased upon downregulation of Suz12 by shRNAs (see "Suz12 shRNA" in **new Fig. 4C**). More importantly, we now use super-resolution confocal microscopy to unequivocally demonstrate co-localization of Suz12 with telomere-binding protein TRF2 (an essential telomere component) (see **new Fig. 4D**). These new results confirm our original data, thus strengthening the main message of the manuscript. We have revised this section in the new manuscript to highlight the functional relevance and the specificity of the co-localization of PRC2 components with telomeres (**page 11, lane 4-22**).

[REVIEWER]: 4) In Figure 3, the recruitment by PRC2 of H3K27me3, H3K9me3, and H4K20me3, and HP1 at telomeres seems poorly supported by the immunofluorescence data, which is notoriously artifact-prone and subjective. Can the authors substantiate this evidence by using molecular

biology approaches that can provide more quantifiable results?

[AUTHORS]: As indicated above, the specificity of the consistent, although low-in-number, interaction of Suz12 with the telomere was confirmed not only by IF with shRNAs and inhibitors against the PRC2 complex but also by ChIP dot-blot and super-resolution confocal microscopy (see current Fig. 5C and new Fig. 5D, and new Fig. 4C,D). Moreover, in the original manuscript, we also show the consequences of this interaction by measuring the levels of the main PRC2's target, H3K27me3 at telomeres by telomeric ChIP dot blot. As seen in the new Fig. 5D the deposition of H3K27me3 at telomeres was dependent on PRC2 levels. In turn, the other heterochromatic marks are affected in the same direction. We have now revised the text to highlight that also this experiment confirms the interaction of Suz12 with the telomere (see page 13, lanes 2-4).

[REVIEWER]: 5) The text reads well, but some minor corrections are needed.

Minor

[REVIEWER]: 1) The authors are encouraged to use consistently TERRA or TERRAs; the latter name seems more accurate, given that the word refers to a collective of TERRA transcripts.

[AUTHORS]: We agree with the reviewer and we have now used only the term TERRAs throughout all the manuscript.

[REVIEWER]: 2) In Figure 1, the same strips at different intensities were used for 'IgG' and 'Input' signals in all 6 panels, was this intended? If it was, the authors should mention this in the legend, as the reader may be confused. Perhaps better, they should display a single stack of strips (all sharing single strips of IgG and input) and add letter numbers to just the graphs. The same layout could be employed in Figure 3 for a similar experiment with shared Input strips.

[AUTHORS]: Although many antibodies share the same IgG, the reason why we organized the display of Figure 1 and 3 in this way is because each antibody requires different exposure times, so each IgG was exposed the very same time for quantification purposes. As suggested by the Reviewer, we explain now in the legend why the same IgG has different exposure times (see page 28, lanes 14-15 and page 30, lanes 19-21).

Detailed answer to reviewer #2

[REVIEWER]: In their manuscript entitled 'TERRA-dependent recruitment of polycomb to telomeres is essential for the assembly of histone trimethylation marks at telomeric heterochromatin' Montero et al. investigate the involvement of the non-coding RNA TERRA in the recruitment of the Polycomb Repressive Complex 2 (PRC2) and its role in the deposition of the heterochromatic histone modifications H3K9me3, H3K27me3, and H4K20me3 at telomeres. This manuscript is an extension of their previous study 'Telomeric RNAs are essential to maintain telomeres' (Montero et al., 2016, Nature Communications 7, 12534) in which the authors identified a subtelomeric region of chromosome 20 (20q) as the source of TERRA RNA in humans. In the present study the authors use a newly derived panel of 20q-deletion cell lines in U2OS cells generated via the CRISPR/Cas9 system. Using these deletion cell lines they find that the 20q-TERRA RNA is required for deposition of the heterochromatic histone marks H3K9me3, H3K27me3 and H4K20me3 at telomeres but not for hypermethylation of subtelomeric DNA. They further show that the Polycomb Repressive Complex 2 (PRC2) binds TERRA RNA and shows localisation at telomeres and that loss of PRC2 leads to a defect in telomeric heterochromatin formation.

I am not a specialist in TERRA or telomere biology and I also have no invested interest in the topic, I can therefore provide an unbiased view. I also start out reviewing scientific manuscripts with a positive attitude and I try to provide constructive feedback. Unfortunately, the work presented in this manuscript has several fundamental flaws and does not meet basic scientific standards. As such I feel that this manuscript is not suitable for publication in Nature Communications or indeed any other scientific journal. I will explain the critical points for this conclusion below

[AUTHORS]: We appreciate that the reviewer has read the previous manuscript in which we identified the 20q as a major TERRA locus in human cells (Montero et al., Nat. Commun., 2016). However, we think that the reviewer did not fully understand the complexity of TERRA and its locus. As described in Montero et al. Nat. Commun., 2016, the 20q-TERRA locus is not the only TERRA locus, which explains many of the concerns raised by reviewer #2, which we have also answered below in detail. The reviewer's misunderstanding is likely due to the fact that he/she is from outside the field as he stated: "I am not a specialist in TERRA or telomere biology".

[REVIEWER]: > The U2OS 20q-TERRA deletion cells that the authors are using in Figure 1 and Figure 2B and 2C are problematic as they are not a genetically 'clean' system. These cell lines were generated using the CRISPR/Cas9 system with 2 guide RNAs that produce a deletion of the subtelomeric 20q locus. While the deletion of the 20q locus in the new panel of clones that the authors use seems to be OK and validated by PCR (Supplementary Figure S1B) there is still substantial expression of TERRA RNA

(Supplementary Figure S1C) in all clones, so clearly this is not a complete TERRA knock out and there are sources of TERRA RNA outside of the 20q locus. The expression of TERRA RNA is very heterogeneous in these cell lines (see Supplementary Figure S1C) which is also reflected in the results, e.g. the authors claim that there is no regulation of telomeric H3K4me3 modification levels while clearly the H3K4me3 levels are ranging from no reduction (clone A7) to only 40% expression compared to wild-type in 50% of the clones (E6 and C4). The same is true for H3K27me3 modification levels, while the authors argue that H3K27me3 levels are generally down-regulated clearly the H3K27me3 levels are only down-regulated in 50% of the clones (clones A7 and E1) but not at all in the other 50% (clones E6 and C4). Such levels of heterogeneity make it very difficult to draw any conclusions and all data based on these 20q-TERRA KO cells have to be treated with extreme caution.

[AUTHORS]: As described in Montero et al., Nat. Commun., 2016, the 20q-TERRA locus is not the only source of human TERRAs **but a major locus, that is why throughout the manuscript we always refer to 20q-TERRA KO cells and not TERRA KO cells.** In the mouse the situation is similar, and TERRAs arise from a main locus but there are other/s that also contribute to TERRA expression (López de Silanes et al., Nat. Commun., 2014). This subtle but important difference explains why we still detect certain amount of TERRA upon deletion of the 20q-TERRA locus. The heterogeneity in TERRA levels in the KO cells lines can be explained both by the clonal expansion and by the growth adaptation of this tumoral cell line. We have revised the manuscript text to clarify these aspects so that readers from outside of the telomere/TERRA field do not get confused (see page 7, lanes 14-16 and page 9, lanes 5-7).

[REVIEWER]: > The main, and principle, problem with the 20q-TERRA 'knock out' cells that the authors have generated is that there is the potential for off-target effects and that these cell lines might contain additional mutations that the authors do not know about and that might cause or contribute to the observed effects. These cells were derived in a similar way as described in the original Montero et al. (2016) study. In this study the authors argue that off-target effects do not constitute a problem since they did not find mutations in 3 of the top predicted off-target loci. However, I do not follow this logic and I think that off-target mutations are a major concern in this case and completely invalidate all experiments using these 20q-TERRA KO cells. Finding no off-target effects in a handful of bioinformatically predicted loci is not an argument for not having off-target mutations elsewhere in the genome where the authors did not check, and therefore off-target mutations cannot be excluded. This is especially problematic in this case where the sole selection criteria for the cells is long-term survival. Telomeres are essential for long term-cell viability and if indeed the 20q-deletion is a TERRA knock out the clones that survive will exactly be those that have gained additional mutations that will help to bypass the problem and allow the cells to survive. The very fact that these cells are viable makes it very likely that the clones that the authors have generated contain additional mutations that compensate for the 20q-deletion and that are somehow involved in telomere regulation processes. The clones they have

isolated therefore have to be regarded as compromised and the results obtained are not trustworthy.

[AUTHORS]: As stated by reviewer #2, we did check for off-targets in the first set of KO clones that we described in Montero et al., Nat. Commun., 2016, among them the C4 KO clone. Indeed, clone C4 has been also used in parallel with the newly generated KO clones in the current manuscript and has been included in the newly revised figures. Importantly, the new KO clones behave similar to the C4 clone in all parameters analysed here including decrease in TERRA levels, decrease in telomere length, and similar impact on the deposition of heterochromatic marks when TERRA levels are reduced. Thus, the similar behaviour of the newly generated 20q-TERRA clones suggesting the absence of off-targets in the new set of 20q-TERRA KO clones. The current and accepted way to analyse off-target effects in the CRISPR-Cas9 field is to analyse the top off-target genes that are predicted bioinformatically, which is the way we did it in Montero et al., Nat. Commun., 2016. Although we cannot exclude the acquisition of natural mutations inherent to the malignant phenotype of the cells used here, the fact that all four different clones used here behave in a similar way in all the parameters analysed strongly supports our findings and therefore we do not agree with the Reviewer's comment.

In the original Montero et al. (2016) study the authors tried to generate 20q-deletion cells from 4 different cell lines (IMR90, HeLa, HCT116 and U2OS). None of the homozygous IMR90, HeLa, and HCT116 20q-deletion clones survived, the only cell line that produced viable clones were U2OS cells and the survival rate of the homozygous U2OS clones was also only 50%. The authors argued that this is due to higher TERRA expression levels in U2OS cells. On the one hand this demonstrates that the 20q-TERRA locus is essential and that the authors are onto something. On the other hand this creates major problems for the present manuscript. Firstly, U2OS cells are obviously outliers and therefore in my opinion not a suitable model to study telomere biology in general. Secondly, upon observing the lethality induced by deleting the 20q-TERRA locus the authors should not have carried out further experiments with these 'promiscuous' U2OS cell lines where results cannot be trusted. Instead they should have established a conditional 20q-TERRA knock out model in a different cell line in which the locus can be inducibly deleted and in which it can be excluded that the cells acquire additional mutations that could influence the results.

In summary and as stated above, any data obtained from the U2OS 20q-TERRA KO cells in this study are invalid as it is not clear whether the observed effects are due to the 20q-TERRA deletion or another unknown mutation. The data presented in the entire Figure 1 and in Figures 2B and 2C can therefore not be used to make the claim that the 20q-TERRA RNA is involved in establishing heterochromatin at telomeres.

[AUTHORS]: Probably owing to the fact that the reviewer is not from the telomere biology field, he/she probably does not know that U2OS cells elongate telomeres not by telomerase but by an alternative mechanism based on homologous recombination termed as Alternative Lengthening of Telomeres or ALT (Dunham et al., Nat. Genet, 2000). Also it is known that ALT cell lines

express high TERRA levels, as we and others also demonstrated before (Nergadze et al., RNA 2009; Flynn et al., Science, 2015; Montero et al., Nat. Commun., 2016). Therefore, we do not agree with the reviewer that U2OS cells are outlayers, instead it is probably the fact that they maintain telomeres in a different manner and that they express high TERRA levels what gives them the advantage to survive with low TERRA levels compared to telomerase-positive cell lines, (we extensively discussed this in Montero et al., 2016. Furthermore, it is of great interest to understand the role of TERRA in these ALT cell lines as they represent a mechanism to maintain telomeres in the telomerase-negative tumors. For instance, it is being shown that ALT renders cancer cells hypersensitive to ATR inhibitors due to altered regulation of TERRA. Thus, the absence of ATRX, such as in U2OS, compromises cell-cycle regulation of TERRA and leads to persistent association of replication protein a (RPA) with telomeres after DNA replication, creating a recombinogenic nucleoprotein structure which is an homologous recombination intermediate but also the responsible of ATR recruitment (Flynn et al., Science, 2005). Moreover, genome-wide targeting of PRC2 depends on ATRX (Sarma et al., Cell, 2014). Since U2OS cells lack ATRX, it is quite possible that both PRC2 localization to telomeres and the interplay between facultative and heterochromatic marks at telomeres in U2OS cells is very different from those in telomerase-positive cells. The persistent TERRA presence at telomeres in ALT cells (Flynn et al., Science, 2005) might be important for PRC2 recruitment in the absence of ATRX. All the former, indicate a distinctive role of TERRA in ALT cells which support our findings. We now discuss these facts in the revised manuscript text (page 16, lines 20-24; page 17, lines 1-3).

[REVIEWER]: > In addition, the immunofluorescence assay to demonstrate telomeric localisation of Suz12 and HP1 is entirely inadequate. Just from looking at the immunofluorescence images it is clear that the stainings for telomere factors (TRF2 in Figures 2B and 3C and Rap1 in Figure 3F) and heterochromatin factors (Suz12 in Figures 2B and 3C and HP1 in Figure 3F) are clearly different and that there is no distinctive localisation of Suz12 and HP1 to telomeres. The number of Suz12 and HP1 fluorescence signals (1.5 for Suz12 and 3 for HP1) per cell cannot be quantified, as I assume the authors have done, like a dotted pattern since clearly both Suz12 and HP1 show diffuse nuclear stainings in the wild-type cells with some regions staining more intensely, as expected. The very low number of co-localisations that the authors report based on their quantifications are likely to result from just chance overlaps of fluorescence signals between the multi-punctuate telomere staining patterns of TRF2 and Rap1 within the diffuse Suz12 and HP1 stainings. The spatial resolution of a normal confocal microscope, such as the Leica SP5-WLL used here, does not allow one to resolve co-localisations at a level that the authors require. The authors can therefore not claim a localisation of Suz12 or HP1 to telomeres, unless they can unambiguously demonstrate and quantify this with approaches that allow identification of bona fide co-localisations at high resolution and with 3D reconstructions. As such the experimental approach presented in Figures 2B and 3C and in Figure 3F is inadequate, and therefore the data are not valid to make the claim that the 20q-TERRA RNA and PRC2 are required for the recruitment of heterochromatin factors to telomeres. Even if the co-localisations were real, around one TRF2-Suz12 (Figures 2B and 3C) or

Rap1-HP1 (Figure 3F) co-localisation event over around 50 telomere spots per cell in wild-type cells, as quantified by the authors, does not indicate extensive telomeric localisation and argues against this phenomenon being a general mechanism.

[AUTHORS]: Although the reviewer is right that few telomeres colocalize with Suz12, we demonstrate that this colocalization is specific as indicated by a significant decrease of in the colocalization of Suz12 with telomeres upon knock-down of either Suz12 or the Suz12-interacting protein Ezh2 by using specific shRNAs, as well as upon treatment with an Ezh2 chemical inhibitor (see Fig. 5). Moreover, we also demonstrate the consequences of this interaction between PRC2 components and telomeres by measuring the activity of the PRC2 complex at telomeres. In particular, we demonstrate presence of H3K27me3 at telomeres by telomeric ChIP dot blot and show that deposition of this mark at telomeres is dependent on PRC2 levels (see new Fig. 5C). Nevertheless, we have now performed additional experiments to strengthen this point. In particular, we have demonstrated the interaction of the PRC2 complex with the telomere by performing a Suz12 ChIP followed by telomeric dot-blot to detect telomeric DNA (see “scramble shRNA” in new Fig. 4C). Furthermore, we show that this interaction is significantly decreased upon downregulation of Suz12 by shRNAs (see “Suz12 shRNA” in new Fig. 4C). More importantly, we now use super-resolution confocal microscopy to unequivocally demonstrate co-localization of Suz12 with telomere-binding protein TRF2 (an essential telomere component) (see new Fig. 4D). These new results confirm our original data, thus strengthening the main message of the manuscript. We have revised this section in the new manuscript to highlight the functional relevance and the specificity of the co-localization of PRC2 components with telomeres (page 11, lane 4-22).

The authors might have a point in demonstrating that the PRC2 complex is involved in establishing telomeric heterochromatin since depletion of the subunits Ezh2 and Suz12 by siRNA leads to loss of HP1, H3K9me3 and H4K20me3 at telomeres, but this has not been investigated further in any detail.

[AUTHORS]: We thank the reviewer for stating that “The authors might have a point in demonstrating that the PRC2 complex is involved in establishing telomeric heterochromatin since depletion of the subunits Ezh2 and Suz12 by siRNA leads to loss of HP1, H3K9me3 and H4K20me3 at telomeres”.

Given the low quality of most of the presented data and that large parts of the manuscript are invalid due to flawed scientific approaches, there is not much valid data that can be taken into account to support the findings claimed by the authors and therefore this manuscript should be rejected.

[AUTHORS]: We hope that the above explanations on the complexity of human TERRAs and their loci, as well as with the addition of new experiments, the reviewer will now value better the interest and relevance of our findings.

Detailed answer to reviewer #3

[REVIEWER]: In this study, Montero and colleagues established additional U2OS clones/lines KO for the 20q-TERRA locus. This is important, given that U2OS cells do not maintain their telomere length within a narrow range and may exhibit large inter-clonal variation. Using these and previously established clones the authors now show that TERRA is partly responsible for the establishment of the H3K9me3 and H4K20me3 constitutive heterochromatin marks (previously known to present at telomeres) as well as a facultative heterochromatin mark H3K27me3 (not known to be present) at telomeres. They further show that 8Å~UUAGGG oligos can interact in vitro with the PRC2 complex subunits Ezh2 and Suz12, which are responsible for H3K27 trimethylation, and that rare instances of PRC2 and TRF2 co-localization are TERRA-dependent. Finally, they provide evidence that PRC2 not only deposits H3K27me3, but also plays a role in accumulation of both H3K9me3 and HP1 at telomeres.

These findings extend our knowledge of the telomeric heterochromatin composition and the way it is established. However, there are a number of caveats and specific comments that must be addressed

[AUTHORS]: We are very grateful to this reviewer for considering that **“these findings extend our knowledge of the telomeric heterochromatin composition and the way it is established”** and for her/his insightful suggestions. In addition, we also thank the reviewer for the detailed review of our manuscript and the very useful commentaries and suggestions, which we have addressed in full in the revised manuscript.

[REVIEWER]: 1. Previously, Maria Blasco’s lab has demonstrated that the majority of TERRA in human cells is transcribed from 20q (Montero et al., Nat Comm 2016). This was confirmed by CRISPR-Cas9-mediated bi-allelic knockout of the putative 20q-TERRA locus, which also allowed to show the importance of TERRA for telomere length maintenance. Curiously, the biallelic deletion of the 20q-TERRA locus was only possible in U2OS cells that maintain their telomeres by ALT, but not in the telomerase-dependent cell lines. The reason for that remains obscure, since U2OS cells that have upregulated TERRA level should be more rather than less sensitive to TERRA KO. In any case, it's important to bear in mind that the studies of TERRA knockout are currently limited to the U2OS cell line that has a very peculiar way of telomere maintenance and therefore it remains unclear whether the findings can be generalized onto telomerase-dependent cells. This point is crucial and the authors must be thoroughly addressed.

[AUTHORS]: Yes, it is probably the fact that U2OS maintain telomeres through ALT and that they express high TERRA levels what gives them the advantage to survive with low TERRA levels compared to telomerase-positive cell lines, (we extensively discussed this in Montero et al., Nat. Commun., 2016). That is

why we think it is of great interest to understand the role of TERRA in these ALT cell lines as it could favor the telomere maintenance in the telomerase-negative tumors. Thus, it is being shown that ALT renders cancer cells hypersensitive to ATR inhibitors due to altered regulation of TERRA. Thus, the absence of ATRX, such as in U2OS, compromises cell-cycle regulation of TERRA and leads to persistent association of replication protein a (RPA) with telomeres after DNA replication, creating a recombinogenic nucleoprotein structure which is an homologous recombination intermediate but also the responsible of ATR recruitment (Flynn et al., Science, 2005). Moreover, genome-wide targeting of PRC2 depends on ATRX (Sarma et al., Cell, 2014). Since U2OS cells lack ATRX, it is quite possible that both PRC2 localization to telomeres and the interplay between facultative and heterochromatic marks at telomeres in U2OS cells is very different from those in telomerase-positive cells. The persistent TERRA presence at telomeres in ALT cells (Flynn et al., Science, 2005) might be important for PRC2 recruitment in the absence of ATRX. All the former, indicate a distinctive role of TERRA in ALT cells which support our findings. We now discuss these facts in the revised manuscript text (**see page 16, lanes 20-24; page 17, lines 1-3**).

[REVIEWER]: 2. In general, the authors confirmed that a large fraction of TERRA is transcribed from the putative 20q-TERRA locus, although analysis of the multiple clones now demonstrated that it's responsible for only 20-50% of human TERRA. This point should be highlighted given discrepancies between the Blasco's Lab and other telomere labs.

[AUTHORS]: The fact that the TERRA downregulation in some 20qTERRA KO clones is only 20-50% might be related to 1) adaptation to the cell culture conditions during clonal cell expansion and 2) compensation of other loci, for example the Xp (Montero et al., 2016). Although we do not exclude the possibility that, in addition to the Xp, other loci could be contributing to TERRA transcription, the 20q (and perhaps the Xp) is the only one in which a formal demonstration of its TERRA genuineness has carried out by genetic means. Nevertheless, in the revised text we now open the possibility to the presence of other loci contributing to TERRA but highlighting that the only one that has been proven so far as a TERRA loci is the 20q-TERRA locus (**see page 7, lane 14-21**).

They also confirmed that 20q TERRA KO results in telomere length reduction: on average KO clones have shorter telomeres compared to controls. However, it's disconcerting that there is no clear correlation between the level of TERRA and telomere length reduction when individual clones are considered (Sup. Figs 1C and 2). It's a pity that the previously established clone C4 that has by far the shortest telomeres (Sup. Fig. 2) has not been analyzed for TERRA level along with the other clones (Sup. Fig. 1).

[AUTHORS]: We did not include the analysis of the C4 because it was already published in Montero et al., 2016 and we did not want to saturate the figures with more panels. However, following the reviewer's suggestion, we have now included analysis of the C4 clone along with the other clones for the RNA-FISH experiments to show total TERRA levels in the C4 clone (**see new Fig. 1**). We

have also revised the text according (see page 8, lanes 1,2 and page 27, lane 15).

[REVIEWER]: 3. Similarly, on average, the 20q-TERRA KO clones have reduced H3K9me3 and H4K20me3 at telomeres (Fig. 1A,B), but there is no correlation between the levels of TERRA (Sup. Fig. 1C,D) and these constitutive heterochromatin marks (Fig. 1A,B). In fact, clone E6 that has the lowest TERRA level (Sup. Fig. 1C) shows the levels of H3K9me3 and H4K20me3 that are the least different from the controls. This lack of "dose-response" relationship between the levels of TERRA and constitutive heterochromatin marks suggests a complex and indirect relationship between the two. The same comment goes for the facultative heterochromatin mark, H3K27me3. How the authors can reconcile this lack of correlation

[AUTHORS]: This apparent lack of "dose-response" might be related to the clonal expansion and growth adaptation of this tumoral cell line. We have now comment and discuss about this issue in the revised manuscript (see page 9, lane 5-9).

[REVIEWER]: 4. The interaction between the biotinylated 8Å~UUAGGG oligos and the Ezh2 and Suz12 subunits of PRC2 in cell lysates (Fig. 2A) is convincing, but this interaction has already been demonstrated in vivo in mESCs using iDRiP-Mass Spec (Chu et al., Cell 2017), which needs to be acknowledged.

A useful control here would be a non-telomeric G-rich oligo to determine whether PRC2 has a general affinity for G-rich RNA especially those capable to fold into G-quadruplex. The effect of 20q-TERRA KO on co-localization between Suz12 and TRF2 (Fig 2B) is less convincing, mainly due to the rarity of these co-localization events (1-2 per nucleus). The number of nuclei analyzed for each clone has to be indicated. This experiment should be strengthened.

[AUTHORS]: the Reviewer is right and we have now acknowledged in the revised manuscript that the demonstration of the direct binding of Ezh2 to TERRA was performed by Chu et al., Cell 2017. Regarding the affinity of PRC2 for G-rich RNA and the use of non-telomeric G-rich oligo as control, all the former was already tested using an EMSA assay by Wang et al., Mol. Cell, 2017. They found that PRC2 has higher affinity for G-rich RNA>C,U>A, especially those capable to fold into G-quadruplex. We have now acknowledged this work in the revised manuscript (see page 10, lanes 19-23; page 11, lanes 1,2)

Although few telomeres colocalize with Suz12, we demonstrate in the manuscript that this colocalization is specific as indicated by the fact that downregulation of either Suz12 or the Suz12-interacting PRC2 protein Ezh2 by shRNA or upon treatment with an Ezh2 chemical inhibitor results in a significant decrease in the colocalizations (see Fig. 5C). Moreover, we also demonstrate the consequences of this interaction between PRC2 components and telomeres by measuring the activity of the PRC2 complex at telomeres. In particular, we demonstrate presence of H3K27me3 at telomeres by telomeric ChIP dot blot and show that deposition of this mark at telomeres is dependent on PRC2

levels (see **new Fig. 5D**). Nevertheless, we have now performed additional experiments to strengthen this point. In particular, we have demonstrated the interaction of the PRC2 complex with the telomere by performing a Suz12 ChIP followed by telomeric dot-blot to detect telomeric DNA (see "scramble shRNA" in **new Fig. 4C**). Furthermore, we show that this interaction is significantly decreased upon downregulation of Suz12 by shRNAs (see "Suz12 shRNA" in **new Fig. 4C**). More importantly, we now use super-resolution confocal microscopy to unequivocally demonstrate co-localization of Suz12 with telomere-binding protein TRF2 (an essential telomere component) (see **new Fig. 4D**). These new results confirm our original data, thus strengthening the main message of the manuscript. We have revised this section in the new manuscript to highlight the functional relevance and the specificity of the co-localization of PRC2 components with telomeres (**page 11, lane 4-22**). We have also included the number of nuclei analyzed in the **new Figure 4B**.

[REVIEWER]: 5. The decrease of the H3K27me3 mark at telomeres in response to either Ezh2 or Suz12 knockdown analyzed by ChIP is very minor, whereas decrease of the H3K9me3 and especially of the H4K20me3 is much more pronounced (Fig 3D). This does not make sense. The IgG background in H3K27me3 ChIP is very high, which perhaps is a problem. To make the data more convincing, it would be nice to repeat the latter. Instead of "recruitment" (line 249) use "deposition" of heterochromatic marks.

[AUTHORS]: As suggested by the Reviewer, we have repeated the H3K27me3 in response to either Ezh2 or Suz12 although the new antibody batch is not working as good as the one we used for the current Fig. 3F. Upon repetition, the quantitative differences between the treatment groups are similar with respect the original H3K27me3 ChIP but the dot-blot image came out clearer and with more contrast that is why we have substituted the original data for the new one (see **new Figure 5D**).

We did not find the word 'recruitment' in line 249. The closer use of 'recruitment' to this line was in line 253 and 254, so we have modified them following the Reviewer's suggestion (see **page 13, lanes 17,19**). If this was not the 'recruitment' you were referring to, please let us know.

[REVIEWER]: 6. Reduction of the HP1 at telomeres in response to either PRC2 knockdown or Ezh2 inhibition analyzed by IF (Fig. 3E) is difficult/impossible to evaluate due to very poor and noisy HP1 signal, but it is quite convincing by ChIP (Fig. 3F). To make a "clean" conclusion regarding the role of PRC2 in HP1 recruitment, the authors should show that Ezh2 knockdown does not result in downregulation of HP1 protein level (by western blot).

[AUTHORS]: Following the Reviewer's suggestion, we have performed a HP1 Western blot upon Ezh2 downregulation. As it can be seen in the **new Suppl. Fig. 1**, HP1 levels are not affected by Ezh2. The text has been revised accordingly (see **page 13, lane 15,16**).

[REVIEWER]: 7. This study provides an additional insight in the composition of the telomeric chromatin (in the ALT cell line), and uncovers the unexpected role

of PRC2 in the accumulation of constitutive heterochromatin marks and HP1 at telomeres. How exactly PRC2 performs this function remains unclear. It's worthy to note that genome-wide targeting of PRC2 depends on ATRX (Sarma et al., Cell 2014). Remarkably, U2OS cells used in this study are ATRX-null. Therefore, it is quite possible that both PRC2 localization to telomeres and the interplay between facultative and heterochromatic marks at telomeres in U2OS cells is very different from those in telomerase-positive cells. The authors may consider mentioning this possibility in the discussion.

[AUTHORS]: We thank the Reviewer for suggesting this idea that we have now included it in the discussion of the revised manuscript (see page 16, lanes 20-24; page 17, lines 1-3).

Reviewers' Comments:

Reviewer #1:

Remarks to the Author:

The authors have addressed my concerns adequately.

Reviewer #2:

Remarks to the Author:

In the revised version of the manuscript "TERRA-dependent recruitment of polycomb to telomeres is essential for the assembly of histone trimethylation marks at telomeric heterochromatin" by Montero et al. the authors have added additional data to support their initial findings and reformulated parts of the manuscript to address the reviewers' comments and to accommodate the new data. I appreciate the difficulties associated with investigating the function of TERRA RNAs and the importance of understanding TERRA function in ALT cells. As stated before, the authors make an interesting point with their findings that TERRA RNAs are involved in recruiting PRC2 and establishing heterochromatin at telomeres. However, in this revised version the authors did not address the fundamental problems associated with this study that I have pointed out in my previous review. It is my honest opinion that any scientific findings must be based on sound experimental evidence. Unfortunately the newly added data and arguments do not alleviate my concerns about the experimental systems used in this study. As I have explained in my previous review the use of the 20q-TERRA knock out U2OS cell lines is problematic as it cannot be excluded that these cell lines have acquired additional compensatory mutations that enables them to survive and thus the observed effects cannot be attributed to the 20q-TERRA KO with certainty. The authors acknowledge this in their rebuttal letter. In general, experiments should be designed in such a way as to exclude ambiguity. After observing the high lethality of the 20q-TERRA KO cells I would have switched to a conditional 20q-TERRA knock out model. The only analysis that I would have undertaken with these cells is to do whole genome, exome and/or RNA sequencing on a sufficiently high number of clones to identify compensatory mutations/deregulations. I do not think that these CRISPR-KO cells are an adequate model to study TERRA function. Also, as detailed in my previous review the immunofluorescence assay to demonstrate overlap between the PRC2 complex, HP1 and telomeres is very weak. The added super-resolution microscopy data does not resolve these problems as the new data has the same issues as the conventional microscopy data and it is still not clear whether the overlap between the PRC2 and telomere stainings is specific or just a chance event.

In summary, I am still of the opinion that the scientific quality of the vast majority of the data presented in this manuscript falls short of what is expected for publication in a journal such as Nature Communications and I cannot recommend acceptance of this manuscript

Additional specific major points:

> Figure 4B: my criticism of the immunofluorescence assay to score for co-localisation of PRC2 to telomeres still stands (see my previous review and next comment for Figure 4D). What does it mean that there is only about 1.5 co-localisations of Suz12 with TRF2 spots? As mentioned in my previous review, a targeting of PRC2 to only 3.5% of telomeres does not argue for a high degree of PRC2 occupancy of telomeres and a general mechanism. Are the telomeres that are targeted by PRC2 somehow special? The authors should show the spread of the number of co-localisations per cell similar to Figure 1D to make the data more transparent.

> Figure 4D: I am not convinced by the newly added super-resolution microscopy data. The representative images shown are not much more convincing or conclusive than the previous ones shown in Figure 4B, I don't see any striking co-localisation between Suz12 and TRF2. How many cells were counted and how many overlaps between Suz12 and TRF1 are there per confocal layer?

What percentage of telomeres are occupied by Suz12? And crucially, what is the probability to get 1 or 2 overlaps between a dotted staining pattern (TRF2) and a diffuse staining pattern (Suz12) just by chance. With such a low number of co-localisations what the authors need to demonstrate is that the 1 or 2 Suz12/TRF2 overlaps per cell/layer are not chance events. Does a random distribution of ca. 50 dots on a diffuse staining such as the one observed for Suz12 in this figure also produce 3.5% overlaps? If yes, the Suz12/TRF2 overlaps are not specific. The same holds true for Figure 5E showing the overlap of the dotted Rap1 staining with the diffuse HP1 staining pattern.

Figure 5C: This control is trivial and only shows that the Suz12 antibody is specific in the IF. If the Suz12 protein is depleted one will lose the IF signal (see left bottom panels) and thereby naturally lose any overlap with another staining pattern, be it specific or non-specific chance overlaps.

Figure 5E: see comments for Figure 4B and 4D.

Minor points:

> The authors should scrutinize the manuscript for grammatical and spelling mistakes.

Reviewer #3:

Remarks to the Author:

The authors fully addressed all our concerns. To this end they performed additional experiments including Suz12 ChIP at telomeres followed by dot-blot, western blot for the HP1 level following PRC2 down-regulation, and even super-resolution microscopy for co-localization of the Suz12 and TRF2.

With all these additional data the main points put forward by the authors become more convincing. The authors also introduced useful explanations of the results and modified the discussion accordingly to our recommendations. Overall, this work sets the stage for investigations into the interplay between telomere transcription and telomeric chromatin function.

We only ask the authors to mention in the abstract that these studies are performed in ALT cells: "Here, we used ALT human cells knock-out for 20q-TERRA to address...."

P.S. We have suggested to use "deposition" instead of "recruitment" with respect to heterochromatic marks (which are deposited and not recruited). This remains to be fixed (see page 13, line 279).

Point-by-point answer to the reviewers

Detailed answer to reviewer #1

[REVIEWER]: Reviewer #1 (Remarks to the Author): The authors have addressed my concerns adequately.

[AUTHORS]: We are delighted to read that this reviewer considers “**the authors have addressed my concerns adequately**”.

Detailed answer to reviewer #3

[REVIEWER]: The authors fully addressed all our concerns. To this end they performed additional experiments including Suz12 ChIP at telomeres followed by dot-blot, western blot for the HP1 level following PRC2 down-regulation, and even super-resolution microscopy for co-localization of the Suz12 and TRF2.

With all these additional data the main points put forward by the authors become more convincing. The authors also introduced useful explanations of the results and modified the discussion accordingly to our recommendations. Overall, this work sets the stage for investigations into the interplay between telomere transcription and telomeric chromatin function.

[AUTHORS]: We are delighted to read that the reviewer thinks that “**The authors fully addressed all our concerns**” and that with the addition of the new data “**the authors become more convincing**”. In particular, regarding Suz12 location at telomeres, the reviewer states “**To this end they performed additional experiments including Suz12 ChIP at telomeres followed by dot-blot, western blot for the HP1 level following PRC2 down-regulation, and even super-resolution microscopy for co-localization of the Suz12 and TRF2**”. Importantly, the reviewer thinks that we have “**set the stage for investigations into the interplay between telomere transcription and telomeric chromatin function**”.

[REVIEWER]: We only ask the authors to mention in the abstract that these studies are performed in ALT cells: "Here, we used ALT human cells knock-out for 20q-TERRA to address...

[AUTHORS]: We have modified the abstract to include the Reviewers suggestion (see page 2, lane 10).

[REVIEWER]: P.S. We have suggested to use “deposition” instead of “recruitment” with respect to heterochromatic marks (which are deposited and not recruited). This remains to be fixed (see page 13, line 279).

[AUTHORS]: We have now used ‘deposition’ instead of recruitment in page 13, line 279 (see page 14, lane 8).

Detailed answer to reviewer #2

[REVIEWER]: In the revised version of the manuscript “TERRA-dependent recruitment of polycomb to telomeres is essential for the assembly of histone trimethylation marks at telomeric heterochromatin” by Montero et al. the authors have added additional data to support their initial findings and reformulated parts of the manuscript to address the reviewers’ comments and to accommodate the new data. I appreciate the difficulties associated with investigating the function of TERRA RNAs and the importance of understanding TERRA function in ALT cells. As stated before, the authors make an interesting point with their findings that TERRA RNAs are involved in recruiting PRC2 and establishing heterochromatin at telomeres. However, in this revised version the authors did not address the fundamental problems associated with this study that I have pointed out in my previous review. It is my honest opinion that any scientific findings must be based on sound experimental evidence. Unfortunately the newly added data and arguments do not alleviate my concerns about the experimental systems used in this study. As I have explained in my previous review the use of the 20q-TERRA knock out U2OS cell lines is problematic as it cannot be excluded that these cell lines have acquired additional compensatory mutations that enables them to survive and thus the observed effects cannot be attributed to the 20q-TERRA KO with certainty. The authors acknowledge this in their rebuttal letter. In general, experiments should be designed in such a way as to exclude ambiguity. After observing the high lethality of the 20q-TERRA KO cells I would have switched to a conditional 20q-TERRA knock out model. The only analysis that I would have undertaken with these cells is to do whole genome, exome and/or RNA sequencing on a sufficiently high number of clones to identify compensatory mutations/deregulations. I do not think that these CRISPR-KO cells are an adequate model to study TERRA function. Also, as detailed in my previous review the immunofluorescence assay to demonstrate overlap between the PRC2 complex, HP1 and telomeres is very weak. The added super-resolution microscopy data does not resolve these problems as the new data has the same issues as the conventional microscopy data and it is still not clear whether the overlap between the PRC2 and telomere stainings is specific or just a chance event

In summary, I am still of the opinion that the scientific quality of the vast majority of the data presented in this manuscript falls short of what is expected for publication in a journal such as Nature Communications and I cannot recommend acceptance of this manuscript

[AUTHORS]: To address the reviewer’s concerns about the use of U2OS but taking into account that different U2OS clones behave in a similar manner (see Fig. 1C, 1D, 2, 3A-F, 4B and 4E), we now acknowledge in the manuscript the limitations of the use of the generated 20q-TERRA knock out U2OS cell lines (see page 2, lane 10; page 9, lane 22-page 10, lane 2; page 17, lane 11-12 and 16; page 18, lanes 20-page 19, lane 2; page 19, lane 11-12).

[REVIEWER]: Additional specific major points:

> Figure 4B: my criticism of the immunofluorescence assay to score for co-localisation of PRC2 to telomeres still stands (see my previous review and next comment for Figure 4D). What does it mean that there is only about 1.5 co-localisations of Suz12 with TRF2 spots? As mentioned in my previous review, a targeting of PRC2 to only 3.5% of telomeres does not argue for a high degree of PRC2 occupancy of telomeres and a general mechanism. Are the telomeres that are targeted by PRC2 somehow special? The authors should show the spread of the number of co-localisations per cell similar to Figure 1D to make the data more transparent.

> Figure 4D: I am not convinced by the newly added super-resolution microscopy data. The representative images shown are not much more convincing or conclusive than the previous ones shown in Figure 4B, I don't see any striking co-localisation between Suz12 and TRF2. How many cells were counted and how many overlaps between Suz12 and TRF1 are there per confocal layer? What percentage of telomeres are occupied by Suz12? And crucially, what is the probability to get 1 or 2 overlaps between a dotted staining pattern (TRF2) and a diffuse staining pattern (Suz12) just by chance. With such a low number of co-localisations what the authors need to demonstrate is that the 1 or 2 Suz12/TRF2 overlaps per cell/layer are not chance events. Does a random distribution of ca. 50 dots on a diffuse staining such as the one observed for Suz12 in this figure also produce 3.5% overlaps? If yes, the Suz12/TRF2 overlaps are not specific. The same holds true for Figure 5E showing the overlap of the dotted Rap1 staining with the diffuse HP1 staining pattern.

Figure 5C: This control is trivial and only shows that the Suz12 antibody is specific in the IF. If the Suz12 protein is depleted one will lose the IF signal (see left bottom panels) and thereby naturally lose any overlap with another staining pattern, be it specific or non-specific chance overlaps.

Figure 5E: see comments for Figure 4B and 4D.

[AUTHORS]: Regarding this reviewer's remaining concerns about the specificity of the interaction of Suz12 and HP1 with the telomere, first we would like to highlight the extensive new experimentation that we included in the reviewed manuscript to address this issue, which we think strongly supports our findings. Importantly, our opinion is also shared by reviewers #1 and #3 (reviewer #3 explicitly states: **"to this end they performed additional experiments including Suz12 ChIP at telomeres followed by dot-blot, western blot for the HP1 level following PRC2 down-regulation, and even super-resolution microscopy for co-localization of the Suz12 and TRF2"**). In particular:

- We have demonstrated the colocalization of Suz12 with the telomere by 3 independent means: confocal microscopy, ChIP dot-blot, and super-resolution microscopy. If a random colocalization between Suz12 and the telomere would have taken place, we could have not gotten specific

signal in the telomeric ChiP dot blot (see Fig. 4C). However, ChiP dot blot clearly shows presence of Suz12 at telomeres (see Fig. 4C).

- Although it is true that the results obtained with the shRNAs only show the specificity of the antibody (which it is important too), the fact that the colocalizations decrease in the TERRA KO cells supports the specificity of the interactions of PRC2 with the telomere in a TERRA-dependent manner (Fig. 4B).
- Moreover, the presence of H3K27me3 at telomere (Fig. 3F and 5D) is another indication of the presence of PRC2 at the telomeres as this enzyme is the only enzyme known to cause this modification.
- Regarding the interaction of HP1 with the telomere, it is a well-established mark at the telomere, which we and others have extensively studied before (see Garcia-Cao, *Nat. Genetics*, 2004). Here, we just confirmed this interaction of HP1 with the telomeres by ChiP dot blot and immunofluorescence (Fig. 4E and 5E) as positive control to prove that this interaction is TERRA-dependent (Fig. 4E).

Nevertheless, we have now performed three different randomization approaches to confirm the interaction of Suz12 and HP1 with the telomere. In particular:

- 1) We have carried out double immunofluorescence of Suz12 and TRF2 (a well-known telomere binding protein) to quantify colocalization of Suz12 with telomeres in parallel with a double immunofluorescence of ACA (that marks centromeres) and TRF2 (that marks telomeres), as a measure of random colocalizations. In a similar manner, we have also included a double immunofluorescence of HP1 and Rap1 (a telomere-binding protein) in parallel with a double immunofluorescence of ACA (centromeres) and Rap1 (telomeres) to measure random colocalizations. Please, note that ACA is an antibody that recognizes the centromere and renders a spotted nuclear pattern upon immunofluorescence (see **new Suppl. Fig. 1**). As it can be seen in the **new Suppl. Fig. 1A and B**, there is a significant increase in the number of colocalizations of Suz12 and HP1 with the telomere binding proteins compared with the ones observed by chance between the telomere binding proteins and ACA. In the representative images, a single confocal layer is shown to avoid any visual artefacts.
- 2) We have also performed an alternative randomization analysis using the interaction plugin on Fiji (Shivanandan et al., *BCM Bioinformatics*, 2013). As it can be seen in the **new Suppl. Fig. 1C and D**, for both co-localizations the strength of the interaction is superior to one, indicating that the spatial distribution of TRF2-Suz12 and Rap1-HP1 are dependent. When compared against 10.000 Monte Carlo samples of point random distributions corresponding to the null hypothesis of “no interaction”, the results are also statistically significant ($p < 0,001$).
- 3) We have also run a randomization approach with the Definiens Developer XD.2 software. We found that the number of random colocalizations of Suz12 with TRF2 and of HP1 with Rap1 was significant lower than in the original pictures. The analysis was run on 14.000 randomized pictures for Suz12-TRF2 and on 4.800 for HP1-Rap1, in

both cases generated from 4 original pictures (**new Suppl. Fig. 1E-L**). The results are also statistically significant ($p < 0,001$ for Suz12-TRF2 and $p < 0,01$ for HP1-Rap1).

Altogether, the three approaches confirm the interaction of Suz12 and HP1 with the telomere. The text has been revised accordingly to include these additional controls (**see page 11, lane 22-page 12-lane 21; page 14, lane 12-page 15, lane 8; page 24, lane 8-22; page 36, lane 2-page 39, lane 13**).

[REVIEWER]: Minor points:

> The authors should scrutinize the manuscript for grammatical and spelling mistakes.

[AUTHORS]: We have revised the manuscript for grammatical and spelling mistakes.